Manuscript prepared for Nonlin. Processes Geophys.
with version 2014/07/29 7.12 Copernicus papers of the LaTeX class copernicus.cls.
Date: 23 September 2016

# Subvisible cirrus clouds – a dynamical system approach

E. J. Spreitzer[1], M. P. Marschalik[1], and P. Spichtinger[1]

[1]Institute for Atmospheric Physics, Johannes Gutenberg University Mainz, Germany

*Correspondence to:* Peter Spichtinger (spichtin@uni-mainz.de)

**Abstract.** Ice clouds, so-called cirrus clouds, occur very frequently in the tropopause region. A special class are subvisible cirrus clouds with an optical depth lower than 0.03. Obviously, the ice crystal number concentration of these clouds is very low. The dominant pathway for the formation of these clouds is not known well. It is often assumed that heterogeneous nucleation on solid aerosol particles is the preferred mechanism although homogeneous freezing of aqueous solution droplets might be possible, since these clouds occur in the low temperature regime $T < 235\,\mathrm{K}$. For investigating subvisible cirrus clouds as formed by homogeneous freezing we develop a simple parcel cloud model from first principles; the model consists of a three dimensional set of ordinary differential equations, and includes the relevant processes as ice nucleation, diffusional growth and sedimentation. We study the formation and evolution of subvisible cirrus clouds in the low temperature regime as driven by slow vertical updraughts $(0 < w \le 0.05\,\mathrm{m\,s^{-1}})$. The model is integrated numerically and also investigated by means of theory of dynamical systems. We found two qualitatively different states for the long-term behaviour of subvisible cirrus clouds. The first state is a point attractor state with a stable focus, i.e. the solution of the differential equations performs damped oscillations and asymptotically reaches a constant value (equilibrium). The second state is a limit cycle in phase space, i.e. the solution approaches asymptotically a state of undamped oscillations. The transition between the states constitutes a Hopf bifurcation and is determined by two parameters – vertical updraughts and temperature. In both cases, the microphysical properties of the simulated clouds agree reasonably well with simulations using a more detailed model, with former analytical studies and with observations of subvisible cirrus. In addition, the model can also be used for explaining complex model simulations close to the bifurcation qualitatively. The results indicate that homogeneous nucleation is a possible formation pathway for subvisible cirrus clouds. The results motivate a mini-

mal model for SVCs, which might be used in future work for the development of parameterisations for coarse large scale models, representing structures of clouds.

## 1 Introduction

Clouds consisting exclusively of ice crystals, so-called cirrus clouds, are frequently found in the tropopause region at low temperatures ($T < 235\,\mathrm{K}$). Satellite observations show frequencies of occurrence up to 40% in extra tropical storm tracks and up to 60% in regions of tropical convection (Stubenrauch et al., 2010). Cirrus clouds influence the energy budget of the Earth-Atmosphere system like other clouds by reflecting and scattering incoming solar radiation (albedo effect) and by absorbing and re-emitting thermal radiation (greenhouse effect). For liquid clouds, the albedo effect usually dominates (Stocker et al., 2013, chapter 7) but for pure ice clouds both effects (albedo vs. greenhouse effect) are of comparable absolute size. Therefore microphysical properties (e.g. size or shape, see Zhang et al., 1999) or macrophysical properties (e.g. optical depth or spatial inhomogeneity) can influence the balance between both radiative effects, leading to a net warming or cooling. Nevertheless, for cirrus clouds a net warming of the Earth-Atmosphere system is often assumed (Chen et al., 2000). Since the formation of ice crystals requires high supersaturation (see, e.g., Koop et al., 2000; Hoose and Möhler, 2012) and diffusional growth of ice crystals is quite slow in the low temperature regime ($T < 235\,\mathrm{K}$) cirrus clouds mostly exist in a thermodynamic state far away from equilibrium. Thus, in contrast to liquid clouds, which approximately coincide with their (super-)saturated environment, for ice clouds there can be a continuous transition from clear air over very low ice crystal number concentrations to thick cirrus clouds with high mass and number concentrations. Cirrus clouds with optical thickness $\tau < 0.03$ constitute a special class, so-called subvisible cirrus clouds (SVCs) (Sassen and Dodd, 1989). These clouds are difficult to measure; remote sensing techniques as LIDAR (e.g., Immler et al., 2008b) or occultation observations (e.g., Wang et al., 1996) are used to detect these very thin cirrus clouds. Only few in situ measurements of subvisible cirrus clouds are available, suggesting very low values in ice crystal number concentrations (Froyd et al., 2010; Kübbeler et al., 2011). Global observations from satellites (Wang et al., 1996; Stubenrauch et al., 2010; Hoareau et al., 2013) as well as observations with stationary LIDAR systems (Sassen and Campbell, 2001; Hoareau et al., 2013) show frequencies of occurrence of about 10–20% in the extra-tropics; in the tropics the frequency of occurrence is much higher (up to 50%, see e.g. Wang et al., 1996). For subvisible clouds, a net warming of the Earth-Atmosphere system is almost certain, since the albedo effect is almost negligible. Our knowledge of subvisible cirrus clouds is quite limited. Since the ice crystal number concentration in SVCs is very low, the question about the dominant formation mechanism is still pending. At cold temperatures ($T < 235\,\mathrm{K}$), where pure ice clouds occur, two different formation mechanisms are generally possible, namely heterogeneous nucleation at solid aerosol particles (e.g. Dufour, 1861; aufm Kampe and Weickmann, 1951; Hosler,

1951) and homogeneous freezing of aqueous solution droplets (Sassen and Dodd, 1989; Koop et al., 2000). For subvisible cirrus, Kärcher and Solomon (1999) stated that both nucleation mechanisms might be possible; in contrast, Jensen et al. (2001) and Froyd et al. (2010) clearly suggested that the dominant mechanism must be heterogeneous nucleation. However, analytical investigations by Kärcher (2002) indicated that also pure homogeneous nucleation might be possible.

In the present study we focus on the formation of SVCs by homogeneous freezing of aqueous solution droplets (hereafter: homogeneous nucleation). We study the formation and evolution of SVCs in an air parcel that is lifted in slow vertical upward motions ($0 < w \le 0.05 \, \mathrm{m \, s^{-1}}$), as typical for synoptic scale motions in the extra-tropics (e.g. along warm fronts, see Kemppi and Sinclair, 2011) or in slow ascent regions in the tropics, as e.g. driven by Kelvin waves (Immler et al., 2008a). We concentrate on the cold temperature regime ($T < 235 \, \mathrm{K}$); thus, we exclude the possibility of liquid origin ice clouds (Krämer et al., 2016; Wernli et al., 2016). This is not a strong limitation since the microphysical properties of ice clouds stemming from mixed phase clouds are quite different, with high ice crystal number and mass concentrations and higher optical depths (Luebke et al., 2016).

For the investigation of subvisible cirrus clouds we develop a parcel model and to which we apply numerical and analytical tools. The model is developed on the basis of an evolution equation for mass distributions of ice crystals, including a description of microphysical processes based on former work (Spichtinger and Gierens, 2009). We take into account the relevant processes for ice microphysics, i.e. ice nucleation, ice crystal growth due to diffusion of water vapour, and sedimentation of ice crystals. For applying analytical tools, we make use of some appropriate simplifications in order to obtain an autonomous system of ordinary differential equations (ODEs); the variables of the system are ice crystal mass and number concentration, respectively, as well as relative humidity with respect to ice. Thus, we have to investigate a three-dimensional autonomous system of ODEs.

To study the qualitative behaviour of the model we use concepts from theory of dynamical systems (see, e.g., Verhulst, 1996; Argyris et al., 2010). For autonomous systems of ODEs, equilibrium states can be found easily. The qualitative properties of the system near the critical points are relevant for the overall behaviour of the system. The stability of these equilibrium states (i.e. point attractors) can be investigated by applying perturbations to the equilibrium states. In fact, we linearise the system at equilibrium points and apply perturbations to this state. The eigenvalues of the linearised system are used for the characterisation of the quality and stability of the equilibrium states. Some theorems are available in order to transfer the qualitative behaviour of the linearised systems to the full nonlinear system. For the characterisation of more complex attractors, as e.g. limit cycles, more sophisticated approaches must be used. For instance, limit cycles can be determined using Poincaré sections (Argyris et al., 2010). Investigations of cloud models as dynamical systems were carried out for liquid and mixed-phase clouds (Hauf, 1993; Wacker, 1992, 1995, 2006) as well as for cloud-aerosol-precipitation systems (Koren and Feingold, 2011; Feingold and Koren, 2013). For pure ice clouds such investigations have not been carried out yet. In contrast to clouds involving liquid phase,

which are close to thermodynamic equilibrium, we have to consider relative humidity as a control

variable, which adds another equation to the system and makes the analysis more challenging. The

mathematical characterisation of the model allows for a better understanding of the interaction of

different nonlinear processes and the impact of external forcings as vertical updraughts. Finally, the

qualitative analysis could be used in future work as starting point for developing cloud parameteri-

sations that represent the qualitative structure of subvisible cirrus clouds.

In section 2 we describe the development of the model. The results of the numerical integration

and the mathematical analysis are presented in section 3, as well as comparisons with observations

and more detailed models. In the final section, we summarise the results, draw some conclusions and

give an outlook to future work.

## 2   Model

In this section we describe the development of a simple model, which is used for analytical and

numerical investigations. We include the relevant processes for formation and evolution of ice clouds

into the model but we try to avoid too much complexity, which makes analysis too complicated (i.e.

reducing the complexity paradox, see, e.g., Oreskes et al., 1994; Oreskes, 2003). Since we investigate

subvisible cirrus clouds in the temperature regime $T < 235\,\mathrm{K}$ and at low vertical updraughts $0 <$

$w \le 0.05\ \mathrm{m\,s^{-1}}$, the relevant processes are ice nucleation, diffusional growth and sedimentation,

respectively.

### 2.1   Basic equations

An ice cloud is represented by an ensemble of ice particles, which can be described by a mass

distribution $f(m,\boldsymbol{x},t)$ with mass of particles, $m$, as internal coordinate and space, $\boldsymbol{x}$, and time, $t$, as

external coordinates. Notation follows the convention in population dynamics (see e.g. Ramkrishna,

2000). We investigate a test volume with a certain fixed mass of dry air, therefore $f$ has units $[f] =$

$\mathrm{kg}^{-2}$. The evolution of this mass distribution in time and space is determined by a partial differential

equation (see, e.g., Hulburt and Katz, 1964; Seifert and Beheng, 2006; Beheng, 2010):

$$\frac{\partial(\rho f)}{\partial t} + \nabla_x \cdot (\rho \boldsymbol{u} f) + \frac{\partial(\rho g f)}{\partial m} = \rho h. \tag{1}$$

Here, $\rho$ denotes density of air, $\boldsymbol{u}$ and $g$ are the advection velocities in physical space and phase

space of the internal coordinate, and $h$ represents sources and sinks for particles. The divergence in

physical space is denoted by $\nabla_x = (\partial/\partial x, \partial/\partial y, \partial/\partial z)$. Note, that all functions, $\boldsymbol{u}$, $g$, $h$, generally

depend on the full set of variables $(m,\boldsymbol{x},t)$. The fluid velocity $\boldsymbol{v} = \boldsymbol{v}(\boldsymbol{x},t)$ describes the motion

of the air; cloud particles may experience a velocity $\boldsymbol{v}' = \boldsymbol{v}'(m,\boldsymbol{x},t)$ relative to $\boldsymbol{v}$, thus the total

$\boldsymbol{u}$ is given by $\boldsymbol{u}(m,\boldsymbol{x},t) = \boldsymbol{v}(\boldsymbol{x},t) + \boldsymbol{v}'(m,\boldsymbol{x},t)$. In our study, the only relevant relative velocity of

cloud particles is gravitational settling (hereafter: sedimentation), given by a terminal velocity due

to balance between gravitational force and drag. The terminal velocity depends on ice crystal mass, i.e. $\boldsymbol{v}' = (0,0,-v_t(m))$. Note the direction towards Earth's surface, indicated by the minus sign.

Instead of solving equation (1) for the entire mass distribution, we derive equations for the general moments of $f(m,\boldsymbol{x},t)$, defined as

$$\mu_k[m](\boldsymbol{x},t) := \int_0^\infty f(m,\boldsymbol{x},t)\, m^k\, \mathrm{d}m, \qquad k \in \mathbb{R}. \tag{2}$$

A bounded mass distribution is uniquely determined by all its integer moments (see e.g. Feller, 1971). However, since we cannot (and do not want to) treat an infinite number of moment equations,

we make the usual ansatz (see e.g. Seifert and Beheng, 2006) for a double moment scheme ($k = 0,1$), i.e. we derive two equations for number concentration ($N_c = \mu_0$) and mass concentration ($q_c = \mu_1$) of ice crystals from equation (1), of the following form:

$$\underbrace{\frac{\partial \mu_k}{\partial t} + \boldsymbol{v}\cdot\nabla_x \mu_k}_{\text{time evolution + advection}} = \underbrace{\frac{1}{\rho}\frac{\partial}{\partial z}\left(\int_0^\infty m^k \rho v_t f\,\mathrm{d}m\right)}_{\text{sedimentation}} + \underbrace{k\int_0^\infty m^{k-1} g f\,\mathrm{d}m}_{\text{growth/evaporation}} + \underbrace{\int_0^\infty m^k h\,\mathrm{d}m}_{\text{particle formation/elimination}}. \tag{3}$$

Here, we also used mass conservation of dry air in order to rewrite the first two terms of equation (1),

for details see appendix A. Note the units of $N_c$ and $q_c$ relative to the mass of dry air are $[N_c] = \mathrm{kg}^{-1}$ and $[q_c] = \mathrm{kg}\,\mathrm{kg}^{-1}$, respectively. For closing the system of equations mathematically, we prescribe a fixed type of mass distribution for the ice crystals. As in the study by Spichtinger and Gierens (2009), we use a log-normal-distribution of the following form:

$$f(m,t) = \frac{N_c(t)}{\sqrt{2\pi}\log\sigma_m}\exp\left(-\frac{1}{2}\left(\frac{\log(\frac{m}{m_m})}{\log\sigma_m}\right)^2\right)\frac{1}{m}, \tag{4}$$

with geometric mean mass $m_m$ and non-dimensional geometric standard deviation $\sigma_m$, determining the width of the distribution; $\log$ denotes the natural logarithm. The general moments can be described by

$$\mu_k[m] = N_c m_m^k \exp\left(\frac{1}{2}\left(k\,\log\sigma_m\right)^2\right) = N_c \overline{m}^k r_0^{\frac{k(k-1)}{2}}, \tag{5}$$

using the mean mass $\overline{m} = q_c/N_c = \mu_1/\mu_0$. Here, we introduced the dimensionless parameter,

$$r_0 = \frac{\mu_2\mu_0}{\mu_1^2} = \exp\left((\log(\sigma_m))^2\right), \tag{6}$$

for closing the set of equations; $r_0$ is set to a constant, thus the geometric standard deviation representing the distribution's width is assumed to be constant. Spichtinger and Gierens (2009) suggest a value of $r_0 = 3$, corresponding to a geometric standard deviation $\sigma_m \approx 2.85$.

## 2.2 Parameterisation of relevant processes

In the following the representation of relevant processes is described briefly. For more details we refer to appendix B. Furthermore, we describe additional assumptions for simplification and present the final equations of the model.

### 2.2.1 Nucleation

For the formation of ice crystals we exclusively consider homogeneous freezing of aqueous solution droplets (short: homogeneous nucleation, Koop, 2004). We describe the ensemble of solution droplets by a size distribution $f_a = f_a(r)$, where $r$ denotes the radius. Units are $[f_a] = \mathrm{kg}^{-1}\mathrm{m}^{-1}$ and $f_a$ is normalised by the total number concentration of solution droplets in dry air, $N_a = \mu_0[r]$.

We model homogeneous nucleation as a stochastic process with a nucleation rate $J$ (for details see appendix B). For the change in the size distribution $f_a(r)$ we can formulate the following equation (acc. to Seifert and Beheng, 2006) assuming $J$ as a volume rate (i.e. $[J] = \mathrm{m}^{-3}\mathrm{s}^{-1}$):

$$\left.\frac{\partial f_a(r)}{\partial t}\right|_{\text{nucleation}} = -\frac{4}{3}\pi r^3 J f_a(r). \tag{7}$$

Integration of the equation leads to an equation for the total loss of solution droplets

$$\frac{\partial N_a}{\partial t} = -\frac{4}{3}\pi \int\limits_0^\infty r^3 J f_a(r)\,\mathrm{d}r. \tag{8}$$

Assuming a bijective relation between ice crystals and solution droplets, we combine the total gain of ice particles as

$$\frac{\partial N_c}{\partial t} = -\frac{\partial N_a}{\partial t} = \frac{4}{3}\pi \int\limits_0^\infty r^3 J f_a(r)\,\mathrm{d}r = \frac{4}{3}\pi J \mu_{3,a}[r], \tag{9}$$

where $\mu_{3,a}[r]$ denotes the third moment of the size distribution of solution droplets. Here, we assume that $\partial J/\partial r = 0$. Since the ice crystal number concentration in SVCs is very low, we assume that only a minor fraction of solution droplets is converted to ice and the size distribution remains constant in time. Thus, the third moment can be calculated once and is then used as a constant in the resulting equations. We assume $f_a(r)$ as a log-normal distribution with a modal radius of $r_m = 100\,\mathrm{nm}$, a dimensionless geometric standard deviation $\sigma_r = 1.5$ and a total number concentration $\rho N_a = 3 \times 10^8 \mathrm{m}^{-3}$, similar to the settings by Spichtinger and Gierens (2009), which are motivated by observations (Minikin et al., 2003). This leads to a formulation of

$$\left.\frac{\partial N_c}{\partial t}\right|_{\text{nucleation}} = \frac{4}{3}\pi N_a r_m^3 \exp\left(\frac{1}{2}\left(3\log\sigma_r\right)^2\right) J(\mathrm{RH}_i, T) \tag{10a}$$

and

$$\left.\frac{\partial q_c}{\partial t}\right|_{\text{nucleation}} = m_0 \cdot \left.\frac{\partial N_c}{\partial t}\right|_{\text{nucleation}}, \tag{10b}$$

using a typical droplet mean mass $m_0 = 10^{-15}$ kg (size $\sim 1\,\mu$m) in the spirit of the mean value theorem. The nucleation rate $J$ is parameterised according to Koop et al. (2000) and can be expressed as a function of relative humidity with respect to ice and temperature. For further details see appendix B.

### 2.2.2 Diffusional growth

The growth and evaporation of ice crystals is dominated by diffusion of water vapour. With several simplifications of the growth equation (for details see appendix B) we obtain the following equation for diffusional growth of a single crystal:

$$g(m) \approx \frac{4}{3} \pi C_i D_v m^{\alpha_i} \rho q_{v,si} (S_i - 1), \tag{11}$$

with constants $C_i = 1.02$ m, $\alpha_i = 0.4$ and using saturation ratio $S_i = p_v / p_{si}$ and saturation mixing ratio

$$q_{v,si}(T,p) = \frac{\varepsilon \, p_{si}(T)}{p}, \tag{12}$$

respectively. We can express the term for diffusional growth in the moment equations (3) by integration, i.e.:

$$\left. \frac{dq_c}{dt} \right|_{\text{growth}} = \int_0^\infty g(m) f(m) \, dm = \frac{4}{3} \pi C_i D_v \rho q_{v,si} (S_i - 1) \mu_{a_i}[m]$$

$$= \frac{4}{3} \pi C_i D_v \rho q_{v,si} (S_i - 1) N_c^{1-\alpha_i} q_c^{\alpha_i} r_0^{\frac{\alpha_i(\alpha_i - 1)}{2}}. \tag{13}$$

### 2.2.3 Sedimentation

Following Spichtinger and Gierens (2009), we describe the weighted terminal velocity $\bar{v}_k$ for the flux of the $k$-th moment as

$$\bar{v}_k = \frac{1}{\mu_k} \int_0^\infty v_t(m) m^k f(m) \, dm, \tag{14}$$

(for details see appendix B). Here, we use a simple power law for the representation of the terminal velocity

$$v_t(m) = \gamma m^\delta \text{corr}(T,p) \tag{15}$$

with $\gamma = 63292.36\,\text{m s}^{-1}\,\text{kg}^{-\delta}$, $\delta = 0.57$ and a density correction term $\text{corr}(T,p)$ (see appendix B). We can compose the general terms for sedimentation in the moment equations (3):

$$\frac{\partial}{\partial z} \left( \rho \bar{v}_n N_c \right) = \frac{\partial}{\partial z} \left( \rho \gamma \cdot \mu_\delta[m] \cdot \text{corr}(T,p) \right), \tag{16a}$$

$$\frac{\partial}{\partial z} \left( \rho \bar{v}_q q_c \right) = \frac{\partial}{\partial z} \left( \rho \gamma \cdot \mu_{\delta+1}[m] \cdot \text{corr}(T,p) \right). \tag{16b}$$

### 2.2.4 Simplifications

In order to obtain a consistent but simplified system of ordinary differential equations we make the following assumptions:

1. Change to Lagrangian point of view and purely vertical motion:

   Since we are interested in the time evolution of cloud variables in a single air parcel, we change our point of view from Eulerian description to a Lagrangian viewpoint. The Eulerian time evolution and advection of a quantity $\phi$ in the fluid motion can be seen as total time derivative

$$\frac{\mathrm{d}\phi}{\mathrm{d}t} = \frac{\partial \phi}{\partial t} + \boldsymbol{v} \cdot \nabla_x \phi, \tag{17}$$

   representing the Lagrangian description. Note that motions relative to the Lagrangian evolution are still included, i.e. sedimentation still plays a role. We will exclusively consider vertical motions of the air parcel as driven by a vertical velocity component $w$, i.e. $\boldsymbol{v} = (0, 0, w(t))$. In order to close the system, equations for temperature and pressure must be derived. The vertical motion of the air parcel leads to adiabatic changes in temperature and pressure. Since we can assume hydro-static balance for pressure in a very good approximation, we explicitly describe temperature and pressure rates:

$$\frac{\mathrm{d}T}{\mathrm{d}t} = \frac{\mathrm{d}T}{\mathrm{d}z}\frac{\mathrm{d}z}{\mathrm{d}t} = -\frac{g \cdot M_{air}}{c_p}w, \qquad\qquad \frac{\mathrm{d}p}{\mathrm{d}t} = \frac{\mathrm{d}p}{\mathrm{d}z}\frac{\mathrm{d}z}{\mathrm{d}t} = -g\rho w. \tag{18}$$

   whereas $g$ denotes acceleration of gravity, $M_{air}$ is the molar mass of dry air and $c_p$ is the molar isobaric heat capacity. We would expect additional temperature changes due to phase changes (latent heat release), when ice crystals grow or evaporate by water vapour diffusion. However, since we investigate ice clouds in the low temperature regime, temperature changes due to latent heat release can be neglected in good approximation. For low temperatures ($T < 235\,\mathrm{K}$) the deviation from the dry adiabatic lapse rate is less than 5% and is decreasing with decreasing temperature Therefore, we omit temperature change due to latent heat release, which would appear as an additional nonlinear term in the system of equations.

2. Closure using an equation for relative humidity w.r.t. ice:

   In our study, we will exclusively consider very low vertical velocities ($0 < w \leq 0.05\,\mathrm{m\,s^{-1}}$), which are typical for formation of SVCs in large-scale upward motions. Variations in $w$, i.e. time-dependent velocities $w(t)$ are not investigated since our main focus is to understand the behaviour of SVCs in this quite simple but realistic setup. Time-dependent vertical velocities would largely complicate our investigations and thus is beyond the scope of this study. At low vertical updraughts, temperature and pressure do not change much. At an updraught velocity of $w = 0.02\,\mathrm{m\,s^{-1}}$, for instance, temperature would decrease by about $0.7$ K per hour. If an updraught of this strength were sustained for $12\,\mathrm{h}$, the resulting temperature decrease would

be about $8\,\mathrm{K}$. A persistence of such weak updraughts for a long time is realistic for warm fronts at mid latitudes (Kemppi and Sinclair, 2011) or Kelvin waves in the tropics (Immler et al., 2008a).

Thus, as temperature decrease at slow upward motions is only very small, in a zeroth order approximation we assume constant temperature and pressure. In consequence, the parcel's volume remains constant, too. The resulting error for neglecting density changes is usually of order $\sim 10\%$ (see e.g. Weigel et al., 2015). Since we are primarily interested in a simple conceptual model describing the main properties of SVCs, these assumptions are justified.

To close the systems of differential equations we introduce an evolution equation for relative humidity, starting with the total derivative of $RH_i = 100\% \, pq_v/(\varepsilon p_{si}(T))$:

$$\frac{\mathrm{d}RH_i}{\mathrm{d}t} = \frac{\partial RH_i}{\partial T}\frac{\mathrm{d}T}{\mathrm{d}t} + \frac{\partial RH_i}{\partial p}\frac{\mathrm{d}p}{\mathrm{d}t} + \frac{\partial RH_i}{\partial q_v}\frac{\mathrm{d}q_v}{\mathrm{d}t}. \tag{19}$$

While temperature and pressure remain approximately the same during parcel ascent, the relative humidity should be affected by terms involving $\mathrm{d}T/\mathrm{d}t$ and $\mathrm{d}p/\mathrm{d}t$, respectively. Neglecting latent heat release as stated above, the first two terms in equation (19) read:

$$\frac{\partial RH_i}{\partial T}\frac{\mathrm{d}T}{\mathrm{d}t} = RH_i \frac{M_{air}}{RT^2} L_{ice} \cdot \frac{g}{c_p}\, w, \tag{20a}$$

$$\frac{\partial RH_i}{\partial p}\frac{\mathrm{d}p}{\mathrm{d}t} = \frac{RH_i}{p} \cdot \rho g w = -RH_i \frac{M_{air}}{RT} g w, \tag{20b}$$

$M_{air}$ is the molar mass of dry air and $L_{ice}$ is the molar heat of sublimation; we use the parameterisation for $L_{ice}$ by Murphy and Koop (2005). As usual, $g$ denotes gravitational acceleration and $c_p$ is the molar isobaric heat capacity of air. Note that we only consider temperature and pressure changes in equation (19), but leave temperature and pressure constant otherwise. Therefore, we do not include the equations for $\mathrm{d}T/\mathrm{d}t$ and $\mathrm{d}p/\mathrm{d}t$ in our ODE system of the model. This approach will be useful for analytical investigations, although this implies a slight inconsistency. This allows us to study the long term behaviour of the system.

The last term in equation (19) represents the sink due to diffusional growth of ice particles and can be written as:

$$\frac{\partial RH_i}{\partial q_v}\frac{\mathrm{d}q_v}{\mathrm{d}t} = -\frac{\partial RH_i}{\partial q_v}\frac{\mathrm{d}q_c}{\mathrm{d}t}\bigg|_{\text{growth}} = -\frac{4}{3}\pi \rho D_v C_i (RH_i - 100\%) r_0^{\frac{\alpha_i(\alpha_i-1)}{2}} N_c^{1-\alpha_i} q_c^{\alpha_i}. \tag{21}$$

We use relative humidity as a control variable instead of specific humidity, which has been used in former studies (e.g. Hauf, 1993; Wacker, 1992) for liquid or mixed-phase clouds close to thermodynamic equilibrium (water saturation). Since pure ice clouds commonly exist at states far away from equilibrium, relative humidity (or equivalently saturation ratio) is the relevant thermodynamic variable. In addition, the representation of processes changing this variable or depending on this variable is much easier than for specific humidity $q_v$, e.g. in the nucleation parameterisation.

3. Approximation of sedimentation

Since we are interested in an analytically treatable model of a single air parcel, we would like to get rid of partial derivatives describing sedimentation, which generally lead to a hyperbolic system of partial differential equations, which is too complicated for theoretical analysis. For simplification of the equations we have to consider terms of the form

$$\frac{\partial}{\partial z}\left(\rho\bar{v}_k\mu_k\right) \qquad k=0,1, \tag{22}$$

i.e. vertical gradients in the sedimentation flux, $j_k = \rho\bar{v}_k\mu_k$. Since the volume does not change, we assume a box with volume $V = A \cdot \Delta z$ with constant vertical extension $\Delta z$ and constant base area $A$. The sedimentation flux $j_k$ is perpendicular to the surface of the base area. We approximate the vertical change of the flux by centred differences:

$$\frac{\partial}{\partial z}j_k \approx \frac{1}{\Delta z}\left(j_k^{\text{top}} - j_k^{\text{bottom}}\right) = \frac{1}{\Delta z}\left((\rho\bar{v}_k\mu_k)^{\text{top}} - (\rho\bar{v}_k\mu_k)_k^{\text{bottom}}\right). \tag{23}$$

We investigate the top layer of a cloud, therefore by definition $j_k^{\text{top}} = 0$. Hence, we can write:

$$\frac{1}{\rho}\frac{\partial}{\partial z}(\rho\bar{v}_N\mu_0) \approx -\frac{\bar{v}_N\mu_0}{\Delta z} = -\gamma\frac{\mu_\delta}{\Delta z}c(T,p), \tag{24a}$$

$$\frac{1}{\rho}\frac{\partial}{\partial z}(\rho\bar{v}_q\mu_1) \approx -\frac{\bar{v}_q\mu_1}{\Delta z} = -\gamma\frac{\mu_{\delta+1}}{\Delta z}c(T,p). \tag{24b}$$

### 2.2.5 Final system of ODEs

In summary, the full system of the model equations reads:

$$\frac{\mathrm{d}N_c}{\mathrm{d}t} = \underbrace{a \cdot J(RH_i,T)}_{\text{nucleation}} \underbrace{-b \cdot N_c^{1-\delta}q_c^{\delta}}_{\text{sedimentation}} \tag{25a}$$

$$\frac{\mathrm{d}q_c}{\mathrm{d}t} = \underbrace{a \cdot m_0 \cdot J(RH_i,T)}_{\text{nucleation}} \underbrace{-c \cdot N_c^{-\delta}q_c^{\delta-1}}_{\text{sedimentation}} + \underbrace{d \cdot (RH_i - 100\%)N_c^{1-\alpha_i}q_c^{\alpha_i}}_{\text{growth}} \tag{25b}$$

$$\frac{\mathrm{d}RH_i}{\mathrm{d}t} = \underbrace{e \cdot w \cdot RH_i}_{\text{vertical motion}} \underbrace{-f \cdot (RH_i - 100\%)N_c^{1-\alpha_i}q_c^{\alpha_i}}_{\text{growth}} \tag{25c}$$

where $a, b, c, d, e, f > 0$ denote positive real constants as indicated in appendix B. Note that almost all coefficients also depend on the (fixed) parameter $T$. This is an autonomous system of ordinary differential equations, i.e. we can write the system in the following form:

$$\dot{\boldsymbol{x}} = \boldsymbol{F}(\boldsymbol{x}), \qquad \text{with } \boldsymbol{x} = (N_c, q_c, RH_i)^T, \tag{26}$$

and $\boldsymbol{F}$ the right hand side of (25). Note that the assumption of constant temperature, pressure and vertical velocity ensures that the system (25) possesses critical points.

### 2.3 Setup

We examine the system for a range of parameter values $0 < w \le 0.05\,\mathrm{m\,s^{-1}}$ and $190\,\mathrm{K} \le T \le 230\,\mathrm{K}$, at a constant pressure of $p = 300$ hPa, which corresponds to upper tropospheric conditions with

moderate vertical motions as in synoptic weather situations or slow upward motions in the tropics
(e.g. Kelvin waves).

We investigate the model using analytical tools (see details in section 3) and also integrate the model numerically. For this purpose, the air parcel is initialised with no ice particles ($N_c(0) = 0$, $q_c(0) = 0$) and at high supersaturation w.r.t. ice ($RH_i(0) = 140$ %). The prognostic equations (25) are integrated numerically with the *LSODA* algorithm from the FORTRAN library *ODEPACK*
(Hindmarsh, 1983).

## 3   Results

### 3.1   General features of the system

The general cloud formation mechanism works as follows: The adiabatic cooling causes the relative humidity, and thus the nucleation rate, to rise until ice nucleation occurs. Due to the steepness of
$J$ with respect to $RH_i$, occurrence of ice nucleation corresponds approximately to a threshold in relative humidity ($\sim 140-150$ %, see, e.g., Ren and Mackenzie, 2005; Kärcher and Lohmann, 2002). The stronger the dynamical forcing $w$, the stronger the nucleation event and the more ice particles form. Ice particle growth then reduces the relative humidity (see equation (19), last term) and hence the nucleation rate is also reduced. Crystals grow to larger sizes and begin to sediment out of the
air parcel. Sedimentation reduces ice crystal mass and number concentrations, and thus weakens the growth term. Then relative humidity can increase again allowing the cycle to start over. The sedimentation process allows for oscillations in the system; without sedimentation (the only sink for $N_c$ and $q_c$) a steady state at ice saturation would be reached soon (see e.g. Kärcher, 2002; Spichtinger and Gierens, 2009).
From the numerical simulations we found that the system exhibits two qualitatively distinct behaviours, depending on values of $w$ and $T$. First, we give a qualitative overview:

**State 1:** At rather high temperatures and slow vertical velocities, the three competing microphysical processes (nucleation, growth, sedimentation) are relatively slow and act on similar time scales, so none of them is dominant. In particular, nucleation rates are rather small in these
cases, therefore only few ice crystals are formed initially, which grow and also sediment quite slowly. The three processes are more or less in balance, resulting in a damped oscillation in all three variables, $N_c$, $q_c$, $RH_i$, asymptotically reaching an equilibrium state, as shown in figure 1. Note, that in this state, nucleation is always present, as strong supersaturation with relative humidity close to the nucleation threshold persists at all times and thus the nucleation
rates are high enough to produce considerable amounts of ice crystals continuously. This results in smooth oscillations instead of sharp nucleation events, as usually expected (see, e.g., Kärcher and Lohmann, 2002). If the air parcel is not disturbed and the vertical updraught remains unchanged in the long term evolution, the cloud persists and has constant microphys-

ical properties. The cloud in the steady state typically contains low crystal concentrations. The dynamic equilibrium remains at high supersaturations, i.e. the cloud stays far away from thermodynamic equilibrium.

**State 2:** When increasing $w$ or decreasing $T$, respectively, to a certain level, oscillations in variables $N_c, q_c, RH_i$ are not damped anymore (see figure 2) and no asymptotic equilibrium can be observed (as e.g. a point in phase space). Instead, we obtain pulse-like nucleation with distinct nucleation events followed by phases with almost vanishing nucleation rates at low relative humidities. The amplitude of the oscillation is very large in all variables; due to sedimentation ice particle concentration is reduced to a small fraction of the maximum value once in a period. At colder temperatures and faster vertical velocities, the nucleation rates are much higher, so nucleation is the dominant process in the beginning, leading to pulse nucleation events. After a while, ice crystal growth becomes dominant and when the crystals have become large, sedimentation sets in and crystal numbers decrease rapidly. Finally, the cycle starts over. In this case, the nucleation events are clearly separated, as opposed to the first case. For the time evolution we find that in the beginning, the amplitude in the three variables decreases slightly from one event to the next, but after a while, the amplitude stays constant. Therefore, it seems that the system asymptotically approaches a limit cycle (one-dimensional attractor). This kind of scenario was also observed in former studies (e.g. Spichtinger and Cziczo, 2010; Kay et al., 2006) but not in a long term behaviour.

Obviously, we find two qualitatively different states in the numerical solution of the model, depending on parameters $w$ and $T$, respectively. Next, we investigate the model by means of qualitative theory of dynamical systems.

### 3.2 Qualitative behaviour of the model

For a first investigation we discuss the different terms in equations (25).

The model is driven by an external source; vertical lifting of the air parcel leads to increase of relative humidity. Since temperature and pressure are kept constant, the term $e \cdot w \cdot RH_i$ implies a permanent water vapour source, which is necessary for studying the long term behaviour of the model. The artificially produced water vapour leads to particle generation. Thus, the terms of nucleation and growth can be seen as internal transformation terms. Finally, sedimentation terms, i.e. $-b \cdot N_c^{1-\delta} q_c^{\delta}$ and $-c \cdot N_c^{-\delta} q_c^{\delta-1}$, remove particles (and thus water mass) from the model, so they constitute internal sinks for cloud variables. Qualitatively, the external sources of water initiate particle generation; diffusional growth terms transform water vapour mass into cloud mass until the mass is lost by the internal sinks of sedimentation. Thus, the model can be seen as an **externally forced dissipative system**. Note, that the model does not fulfil mass conservation due to the sources and sinks of water

vapour and cloud mass, respectively. All terms except of the cooling term $e \cdot w \cdot RH_i$ are non-linear in variables $N_c, q_c, RH_i$.

For a first analysis we compute the divergence of the system (i.e. the trace of the Jacobian $\mathbf{DF}$):

$$\nabla \cdot \boldsymbol{F} = -\left[ (b(1-\delta) + c(1+\delta))N_c^{-\delta} q_c^{\delta} + f N_c^{1-\alpha_i} q_c^{\alpha_i} \right] + e \cdot w + d\alpha_i (RH_i - 100\%) N_c^{1-\alpha_i} q_c^{\alpha_i - 1}$$

$$= -\left[ (b(1-\delta) + c(1+\delta))\overline{m}^{\delta} + f N_c \overline{m}^{\alpha_i} \right] + e \cdot w + d\alpha_i (RH_i - 100\%)\overline{m}^{\alpha_i - 1} \tag{27}$$

using the mean mass $\overline{m} = q_c / N_c$ for cloudy states. For clear air ($N_c = q_c = 0$), we obtain $\nabla \cdot \boldsymbol{F} = e \cdot w > 0$, hence the system is expanding in phase space. For cloudy air ($\overline{m} \neq 0$) there is competition between different terms determining the sign of $\nabla \cdot \boldsymbol{F}$. Sedimentation and change of relative humidity due to diffusional growth are dissipative terms (i.e. negative sign in equation (27)), while the external source term always has a positive sign. Diffusional growth of ice particles can change its sign depending on the thermodynamic state. Since we always investigate a situation with $w > 0$, the system stays in a supersaturated states ($RH_i - 100\% > 0$), therefore the last term in equation (27) is positive.

The balance of terms in equation (27), i.e. the sign of $\nabla \cdot \boldsymbol{F}$ for cloudy air is crucially determined by the mean mass of the cloud. Note that for both exponents we have $0 < \alpha_i < \delta < 1$, and thus $-1 < \alpha_i - 1 < 0$. For large ice crystal mass, the terms of form $\overline{m}^{\delta}$ will dominate, thus leading to a negative sign of $\nabla \cdot \boldsymbol{F}$ and to dissipation of the system, mainly due to sedimentation of ice crystals. This is especially the case at higher temperatures, since then diffusional growth is faster and mean masses $\overline{m}$ tend to larger values. In such cases, the system tends to state 1.

For very small ice crystals, the term including $\overline{m}^{\alpha_i - 1}$ will dominate leading to a positive sign of $\nabla \cdot \boldsymbol{F}$. For instance, at nucleation events, the ice crystal mass becomes very small, thus in this situation the system tends to expand explosively ($\nabla \cdot \boldsymbol{F} > 0$). The same is true if almost all particles have fallen out and only small ice crystals are contained in the air parcel. These scenarios are more prevalent at state 2, i.e. at lower temperatures and higher upward velocities.

### 3.3 Linear stability of the system

In a first step, the dynamical system (25) can be characterised by its critical points $\boldsymbol{x_0}$, i.e. the points in phase space where $\boldsymbol{F}(\boldsymbol{x_0}) = \boldsymbol{0}$. Since the system is autonomous the critical or singular points are equilibrium states of the system. The equilibrium points of this system cannot be determined analytically, due to strong nonlinearities. We determine the roots of the right hand side of system (25) numerically. First, we observe that the mass rate of nucleation $\frac{dq_c}{dt}\big|_{\text{nucleation}} = a \cdot m_0 \cdot J(RH_i, T)$ is negligible compared to other mass rates in the system and can be omitted for simplification. This leads to a new system $\dot{\boldsymbol{x}} = \tilde{\boldsymbol{F}}(\boldsymbol{x})$. After setting $\tilde{\boldsymbol{F}}(\boldsymbol{x}) = 0$, the three resulting equations can be combined to a single equation for $RH_i$ as follows:

$$a \cdot J(RH_i, T) = \frac{e \cdot w \cdot b}{f} \cdot \left( \frac{d}{c} \right)^{\frac{\delta - \alpha_i}{\delta + 1 - \alpha_i}} \cdot RH_i \cdot (RH_i - 100\%)^{\frac{1}{\alpha_i - 1 + \delta}}. \tag{28}$$

For details of the derivation of this equation see appendix C. The roots of equation ((28)) determine the equilibrium values of $RH_i$. Then, the values of $N_c$ and $q_c$ can be derived analytically. Equation (28) has a unique solution because the left-hand side is a strictly monotonic increasing function of $RH_i$ and the right-hand side is strictly monotonic decreasing. Therefore, there exists a unique critical point, $\boldsymbol{x_0}$, in the relevant domain of the phase space ($RH_i > 100~\%$, $N_c > 0$, $q_c > 0$). The roots of equation (28) are determined numerically for the relevant domain in the parameter space, i.e. $0 < w \leq 0.05~\mathrm{m\,s^{-1}}$ and $190 \leq T \leq 235$ K.

In order to examine the qualitative behaviour of the solution in a neighbourhood of the equilibrium state, the ODE system is linearised about the critical point $\boldsymbol{x}_0$:

$$\dot{\boldsymbol{x}} = \boldsymbol{F}(\boldsymbol{x_0}) + \mathbf{DF}\big|_{\boldsymbol{x}_0}(\boldsymbol{x} - \boldsymbol{x}_0) + \mathcal{O}(|\boldsymbol{x} - \boldsymbol{x}_0|^2), \tag{29}$$

where $\mathbf{DF}|_{\boldsymbol{x}_0}$ is the Jacobian of $\boldsymbol{F}$ evaluated at $\boldsymbol{x}_0$. Note that $\boldsymbol{F}(\boldsymbol{x}_0) = 0$ by definition. The three eigenvalues of the Jacobian, $\lambda_1, \lambda_2, \lambda_3$, determine the quality of the critical point (Verhulst, 1996, Chapter 3). The eigenvalues must be determined numerically for the relevant parameter values $w$ and $T$. The Jacobian of the system has two complex conjugate eigenvalues, $\lambda_{1,2} \in \mathbb{C}$, whose real part can be positive or negative, depending on the parameters, $w$ and $T$. In figure 3 the values of the real part $\mathrm{Re}(\lambda_{1,2})$ and the absolute value of the imaginary part $|\mathrm{Im}(\lambda_{1,2})|$ are shown. The third eigenvalue, $\lambda_3 \in \mathbb{R}$, is always negative, values are shown in figure 4.

Complex eigenvalues of the linearised system indicate oscillatory behaviour, which is prevalent in all simulations. As can be seen in figure 3, the real part of the complex eigenvalues $\lambda_{1,2}$ can change its sign depending on parameters $w$ and $T$.

For negative values of the real part ($\mathrm{Re}(\lambda_{1,2}) < 0$) the critical point $\boldsymbol{x}_0$ is a positive attractor, i.e. solutions of the ODE (29) starting in a neighbourhood of this point approach this point asymptotically (Verhulst, 1996, Chapter 2). More precisely, this equilibrium point can be characterised as stable focus (e.g. Verhulst, 1996; Argyris et al., 2010). According to the Poincaré-Lyapunov theorem (Verhulst, 1996, theorem 7.1), positive attraction in the linearised system is also valid for the full nonlinear system (25). Therefore, $\boldsymbol{x}_0$ is asymptotically stable and acts as a positive point attractor in equation (25).

This equilibrium point (stable focus) corresponds to state 1 in the numerical simulations. Solutions of the system (25) experience damped oscillations until they asymptotically approach the stable attractor in phase space. The imaginary part of the complex eigenvalues determines the oscillation period. Figure 5 shows the trajectory of a solution of the system (25) in the 3D phase space, spiralling towards the equilibrium point, i.e. the positive attractor.

For positive values of the real part ($\mathrm{Re}(\lambda_{1,2}) > 0$) the critical point $\boldsymbol{x}_0$ is a negative point attractor (unstable focus). Solutions starting in a neighbourhood of $\boldsymbol{x}_0$ run away from the unstable equilibrium point. In this case, the characterisation of an unstable critical point in the linearised system is not sufficient for a general characterisation of the full nonlinear system, since after short time the solutions are too far away from the equilibrium points. Numerical integration shows undamped oscillations

for solutions that do not start in the equilibrium point; this behaviour points to the possibility of a limit cycle (one-dimensional attractor). The transition from positive point attractor to limit cycle is a so called Hopf bifurcation (Verhulst, 1996) and is associated with a transition from two conjugate complex eigenvalues with negative real part to two conjugate complex eigenvalues with positive real part, via two purely imaginary eigenvalues. For vanishing real part of $\lambda_{1,2}$, the Hopf bifurcation

occurs. The existence of a limit cycle cannot be shown analytically for this system; however, we can determine the limit cycle numerically. For this purpose, we compute the Poincaré map of the system (Argyris et al., 2010; Verhulst, 1996). We choose a two-dimensional plane $\Sigma$ in phase space, which is transverse to the trajectory of the solution of equation (26); $\Sigma$ is called Poincaré section. The sequence of points in phase space where the trajectory crosses $\Sigma$ converges numerically to the

the point on the limit cycle that is in $\Sigma$. Once we find one such a point on the limit cycle, we can use it as the initial condition in (26) to compute the complete limit cycle. An example of a Poincaré section for determining the respective limit cycle is shown in appendix D (figure 16). The limit cycle itself constitutes a one-dimensional positive attractor, i.e. solutions starting outside of the limit cycle approach the limit cycle asymptotically. Figure 6 shows the trajectory of a solution of the system

(25) in the 3D phase space, approaching the limit cycle, which constitutes a warped circle in phase space.

   The transition between the two general states of the system (stable point attractor vs. limit cycle) can be represented in a bifurcation diagram of the $w$-$T$-space (figure 7). The bifurcation point is a function of both $w$ and $T$. The different states are separated by points with vanishing real part of

eigenvalues $\lambda_{1,2}$, indicated by the thick black line. The bifurcation points were obtained numerically.

## 3.4 Quantitative overview

After discussing the different states of the system qualitatively, we now give an overview of the quantitative cloud properties and relative humidity for the point attractor and the limit cycle, respectively.

In the "point attractor" regime (stable focus), i.e. **state 1** of the system, the critical point corresponds to the equilibrium values within the finally persisting cloud. Hence, in this parameter regime, we describe the properties of the modeled cloud by the values of the system variables at the critical point. For the "limit cycle" regime, i.e. **state 2** of the system, the critical point does not describe the changing properties of the cloud since it is only in the centre of the periodic orbit and the trajectory

does not approach it. A more revealing measure for the cloud properties in this regime is a probability density of the values the variables take along the limit cycle, or at least median, maximum and minimum values.

   Figure 8 shows ice crystal mass and number concentrations, respectively, at the critical point, $\boldsymbol{x_0}$, as a function of vertical velocity ($q_c(w)$, $N_c(w)$) for different temperature regimes. The solid

lines in both panels correspond to state 1 (point attractor regime, damped oscillations), whereas the

dashed lines indicate the values at the critical point, $x_0$, for state 2 (limit cycle regime, undamped oscillations); note that for state 2, $x_0$ is an unstable focus.

Ice crystal number concentrations at the critical point take values in the range $3 \times 10^2 \text{ kg}^{-1} \leq N_c \leq 2 \times 10^5 \text{ kg}^{-1}$ (figure 8, top), which corresponds to ice crystal number densities of $0.1 \text{ L}^{-1} \leq n_c \leq 110 \text{ L}^{-1}$. Ice crystal mass concentration ranges between $4 \times 10^{-9} \leq q_c \leq 3 \times 10^{-6} \text{ kg kg}^{-1}$ (figure 8, bottom). This corresponds to an ice water content of $2.2 \times 10^{-9} \leq IWC \leq 1.4 \times 10^{-6} \text{ kg m}^{-3}$.

As expected from theory (e.g. Kärcher and Lohmann, 2002) and from former numerical investigations (e.g. Spichtinger and Gierens, 2009), the ice crystal number concentrations display a strong increase with rising vertical velocity. Due to increased crystal growth rates at higher temperatures, $N_c$ decreases with rising $T$. In the double logarithmic representation in figure 8, the number concentrations $N_c(w)$ at $x_0$ appear as straight lines. For different temperature regimes, there seems to be a constant shift between the curves $N_c(w)$, leading to parallel lines in the double logarithmic representation.

For the limit cycle regime (state 2), we can still derive the values of mass and number concentrations at the critical point, $x_0$. However, since this point is an unstable focus, another representation is needed to describe the range of ice crystal concentrations. As indicated in figures 7 and 8, the limit cycle behaviour occurs for temperatures $T < 230 \text{ K}$ for the investigated updraught regime $0 \leq w \leq 0.05 \text{ ms}^{-1}$. Thus, in figure 9 we present maximum and minimum values (dashed lines) and median values (dot-dashed lines) for ice crystal number concentrations in the limit cycle regime for temperatures $T = 190, 200, 210, 220 \text{ K}$. In addition, the ice crystal number concentration at the critical point, $x_0$, is displayed (solid lines). We observe a large variation in the number concentrations of up to two orders of magnitude relative to the median. This behaviour is reasonable since sedimentation reduces the amount of ice crystals in a dominant manner, while new ice crystals are formed by nucleation in a pulsating way. The absolute values are in the range $0.2 \leq n_c \leq 200 \text{ L}^{-1}$.

The mass concentration of the ice crystals is largely determined by the efficiency of diffusional growth. As indicated in the model description (section 2), this term depends on temperature and also on number concentration, leading again to a power law relationship as represented in figure 8 (bottom) and to a constant shift between the different temperatures, represented as parallel lines.

For the point attractor regime, we can directly investigate the mean mass of the ice crystals, $\overline{m} = q_c/N_c$, at $x_0$, which is displayed in figure 10. The variation of $\overline{m}$ at the critical point due to the vertical velocity is marginal, as indicated in the figure. Thus, we can assume that $\overline{m}$ can be approximated by a function of temperature. The mean mass at $x_0$ ranges between $\overline{m} \sim 10^{-12} \text{ kg}$ and $\overline{m} \sim 2 \times 10^{-10} \text{ kg}$, which corresponds to mean sizes between $\overline{L} \sim 16 \, \mu\text{m}$ and $\overline{L} \sim 134 \, \mu\text{m}$. For the limit cycle regime (state 2), we indicate the variation in the mean mass by box and whiskers plots, displaying the median value (red markers) as well as $25/75\%$ percentiles and minimum/maximum values. Note here that variation of mean mass is usually of one order of magnitude. For cold temper-

atures the variation is larger due to a higher variability in ice crystal number concentration (see figure 9), whereas the mass concentration in ice clouds is mainly dominated by available water vapour.

As indicated in section 3.3, the imaginary part of the complex eigenvalues $\lambda_{1,2}$ determines the period of the oscillations in state 1 near the equilibrium point. In figure 11 the period $\tau = \frac{2\pi}{\text{Im}(\lambda_{1,2})}$ as calculated from the imaginary part is shown for the stable focus (solid lines, colours indicate different temperature regimes). For the unstable focus, the imaginary part of the eigenvalues is not meaningful, as the limit cycle is not within the linear regime of $x_0$. Instead, the periods of the limit cycle is shown (dashed lines, colours indicate different temperature regimes) as calculated from the Poincaré map. Note that for decreasing temperature the period $\tau$ becomes very large.

### 3.5 Comparison with observations

For comparison with observations we first consider in situ measurements of ice crystals in subvisible cirrus clouds. Since it is very difficult to measure low number concentrations, only few measurement studies are available. We compare our results with measurements by Kübbeler et al. (2011), Lawson et al. (2008) and Davis et al. (2010).

Our model results lead to ice crystal number concentrations in the range $0.1\,\text{L}^{-1} \leq \rho N_c \leq 200\,\text{L}^{-1}$ and mean ice crystal sizes in the range $\sim 16\,\mu\text{m} \leq \overline{L} \leq 134\,\mu\text{m}$. Note, that the variation in number concentrations span over three orders of magnitude and the variation in mean sizes is still within two orders of magnitude. These values agree quite well with the measurements. Kübbeler et al. (2011) observed quite high number concentrations in order of $\sim 100\,\text{L}^{-1}$ for small ice crystals ($L \sim 10\,\mu\text{m}$) but quite low number concentrations $0.1 \leq \rho N_c \leq 10\,\text{L}^{-1}$ for large ice crystals (equivalent radius $r > 50\,\mu\text{m}$). Lawson et al. (2008) reported ice crystal number concentrations in the range $22.5 \leq \rho N_c \leq 188.8\,\text{L}^{-1}$ with mean value and standard deviation $66 \pm 30.8\,\text{L}^{-1}$ for ice crystals in the size range $1 \leq L \leq 200\,\mu\text{m}$. Finally, Davis et al. (2010) reported very low ice crystal number concentrations with a mean value of $2\,\text{L}^{-1}$ and mean sizes of $14\,\mu\text{m}$ during the tropical measurement campaign TC4. However, in their study values from former measurement campaigns are reported to be in the range $10 \leq \rho N_c \leq 100\,\text{L}^{-1}$ and for effective radii $10 \leq r \leq 20\,\mu\text{m}$. In a second step we expand our comparison to observations from remote sensing. Since SVCs are optically very thin, we investigate the extinction coefficient for the visible part of the spectrum. For comparing our results with measurements, we calculate the extinction $\beta$ in the solar range using parameterisations by Fu and Liou (1993):

$$\beta = IWC \cdot \left( a + \frac{b}{D_e} \right), \tag{30}$$

where $IWC = q_c \cdot \rho$ denotes ice water content in $\text{g\,m}^{-3}$ and $D_e$ is the generalised size. Constants are given by $a = -6.656 \cdot 10^{-3}\,\text{m}^2\text{g}^{-1}$ and $b = 3.686\,\mu\text{m\,m}^2\text{g}^{-1}$. As a useful approximation we set $D_e = \overline{L}$, where the quantity $\overline{L}$ is calculated from the mean mass $\overline{m}$ using the mass-length-relation $\overline{L} = C_i \overline{m}^{\alpha_i}$, as indicated in appendix B. In figure 12 the values for $\beta$ are shown for different temper-

ature regimes as calculated for the mean values at the (stable and unstable) focus (equilibrium point). Note that there is only marginal difference in the values for different temperatures. The values are within the interval $10^{-4} \leq \beta \leq 0.02 \text{ km}^{-1}$.

Seifert et al. (2007) report mean values for extinctions of SVCs in the range $0.015 \leq \beta \leq 0.02 \text{ km}^{-1}$ with standard deviations $\sigma \sim 0.005 - 0.009 \text{ km}^{-1}$ (see their table 3). Our results are in the same order of magnitude or even smaller for slow vertical updraughts. Davis et al. (2010) report much smaller values of extinction scattered in the range $0 < \beta < 0.01$ with a mean value of $\overline{\beta} \sim 0.001 \text{ km}^{-1}$. These SVCs were measured in the tropics at high altitudes ($z \sim 16$ km), i.e. at low temperatures

$T < 195 \text{K}$, where slow large-scale updraughts due to Kelvin waves in order of $w < 0.01 \text{ m s}^{-1}$ dominate (Immler et al., 2008b). This is consistent with our results, see figure 12.

    Overall, we can state that regarding the high spread in the measurements our results from a simple analytical model agree quite well with in situ measurements.

### 3.6   Comparison with other models

For comparison with a more detailed model we carried out simulations with the box-model described by Spichtinger & Gierens (2009) and Spichtinger & Cziczo (2010). This model includes more sophisticated treatment of microphysical processes, although it is also a two-moment bulk model. It allows a change in the shape of ice crystals from almost spherical droxtals to columns. Homogeneous nucleation is treated in detail, including deliquescence of sulphuric acid and integration over

the full size distribution of solution droplets. For diffusional growth, kinetic and ventilation effects are included. Finally, temperature and pressure changes due to vertical upward motions and latent heat release is added to the air parcel's temperature.

    Henceforth this model is termed "complex model". We scan through the $T$-$w$ parameter space using initial temperatures in the range $190 \leq T \leq 235 \text{K}$ with a temperature increment of $\Delta T = 5$ K

and vertical velocities in the range $0.005 \leq w \leq 0.05 \text{ m s}^{-1}$ with a velocity increment of $\Delta w = 0.005 \text{ m s}^{-1}$, leading to 90 simulations. Additionally, we fixed initial conditions $p = 300 \text{ hPa}$ and $RH_i = 140\%$. Generally, the results of these simulations are in good agreement with the results of the analytical model.

    We can again identify regimes in the $T$-$w$ parameter space showing the known two different

states, i.e. damped oscillations (state 1) and limit cycle behaviour (state 2). In figure 13 the case of damped oscillation is shown in both model simulations. Here, initial temperature of $T = 220 \text{K}$ is used with a vertical velocity of $w = 0.01 \text{ m s}^{-1}$. Green lines indicate the evolution in the complex model simulation, whereas blue lines represent the evolution in the simple analytical model. For the variables number and mass concentration, both models produce almost the same values. The onset

of ice nucleation is shifted between the two models due to differently detailed representation of ice nucleation in both models. This leads to the difference in relative humidity values. Qualitatively, the models agree very well – the oscillation periods and the magnitudes of the damping are very similar.

For the complex model simulations the environmental conditions change, i.e. temperature and pressure are decreasing due to adiabatic expansion. Thus, no steady state can be reached. The values for ice crystal number concentrations and relative humidity are slightly rising with time in the quasi steady state at the end of the simulation. Ice crystal mass concentration is slightly decreasing.

In figure 14, a case of limit cycle behaviour is shown. As in figure 13, green lines indicate the complex model simulations and the simple model results are represented by blue lines, respectively. The initial conditions for both models are given by $T = 210\,\mathrm{K}$ and $w = 0.02\,\mathrm{m\,s^{-1}}$. Again, we find very good agreement in the cloud variables $N_c$, $q_c$ between both model simulations. Qualitatively they also agree very well in terms of the periods of the oscillations.

The bifurcation diagram displayed in figure 7 cannot be reproduced accurately by the complex simulations. Since in the complex model the parameter $T$ is changed during the simulations, switching from one regime to the other is possible within one simulation. If, for instance, a simulation starts at a point in parameter space within the point attractor regime (e.g. high temperature at low updraughts), the time evolution initially follows the damped oscillations as expected from the bifurcation diagram of the simple model. However, the temperature change leads to a (horizontal) path in the phase diagram (figure 7) and at some stage the boundary between the two states is crossed, and from then on, the system will perform undamped oscillations. Indeed, we observe this transition in the complex model simulations. An example for this situation is given in figure 15, with initial conditions $T = 225\,\mathrm{K}$ and $w = 0.035\,\mathrm{m\,s^{-1}}$. Note that in the limit cycle regime the properties of the theoretically expected limit cycle also change with decreasing $T$. This results in increasing amplitudes of the oscillations in $N_c$, $q_c$, $RH_i$ and in increasing periods. Thus, we can conclude that for realistic simulations including changes in environmental conditions there could be transitions between the theoretically determined states. However, the behaviour of the actual states can still be explained by the phase diagram as obtained from our analytical considerations.

We also compare our results with the analytical model Kärcher (2002). This model includes a more sophisticated representation of nucleation and growth. The relevant equations are treated using typical time scales and approximation of the occuring intergrals. Comparison with theoretical results by Kärcher (2002) shows good agreement as well. Actually, in our investigations with the simple analytical model we found low ice crystal number concentrations similar to results by Kärcher (2002); the dependence of number concentrations on $w$ and $T$ also agrees very well with analytical considerations by Kärcher (2002). However, our approach goes beyond the results by Kärcher (2002) since we allow for sedimentation of ice crystals. This additional process leads to the oscillatory behaviour in both states, whereas in the study by Kärcher (2002) a steady state at ice saturation is reached soon. Especially the continuous nucleation in the state 1 scenario (damped oscillation) is only possible if we allow for sedimentation of ice crystals. Otherwise, the nucleation event would stop after depositional growth has reduced the supersaturation such that nucleation rates become negligible. Thus,

we can state that our scenarios might be more realistic, although the microphysical properties in both

studies are quite similar.

## 4    Conclusions

In this study we developed an analytical model for describing subvisible cirrus clouds formed by homogeneous nucleation in the tropopause region. The model consists of a set of autonomous ordinary differential equations for the variables ice crystal mass and number concentration, and relative hu-

midity with respect to ice. It contains the relevant cloud processes ice nucleation, diffusional growth and sedimentation. The model can be viewed as an externally forced dissipative system. The model is integrated numerically and also investigated using linear theory of dynamical systems.

Integration and theoretical analysis show that the system contains two different states, a point attractor state and a limit cycle state. The states depend on the environmental parameters vertical

updraught, $w$, and temperature, $T$. The transition between the states can be described as Hopf bifurcation. Both states show oscillatory behaviour, either damped (point attractor) or basically undamped (limit cycle).

The microphysical properties of the cloud in both states depend mostly on the environmental conditions as vertical velocity and temperature. However, for the limit cycle case the spread in ice

crystal mass and number concentration is obviously larger than in the attractor case. For the stable point attractor, the mean mass depends only slightly on vertical velocity, thus we can approximate the mean mass as a function of temperature.

Comparisons with a more detailed box-model by Spichtinger and Gierens (2009) show very good agreement. The qualitative behaviour as determined for the analytical model can also be found for

the complex model simulations. Also, in terms of quantitative results both models agree quite well. Former analytical investigations by Kärcher (2002) show good agreement with our model, too. However, since we include sedimentation in our model, our results go clearly beyond the former investigations; the long-term behaviour is different, since the inclusion of sedimentation crucially leads to the bifurcation, depending on environmental conditions.

Since there are only few in situ measurements of subvisible cirrus available, it is quite difficult to carry out solid comparisons. However, we try to compare with measurements as described by Kübbeler et al. (2011), Lawson et al. (2008), and Davis et al. (2010) and find good agreement with our results. Also the extinction coefficient as calculated from model results agree very well with observations obtained with remote sensing techniques (Seifert et al., 2007; Davis et al., 2010).

The major qualitative results can be summarised as follows:

- We could show that homogeneous freezing of aqueous solution droplets at low temperatures ($T < 235\,\mathrm{K}$) is a possible pathway for the formation of subvisible cirrus clouds at low vertical

updraughts. Thus, the question about the dominance of formation mechanisms for these thin clouds remains open (homogeneous vs. heterogeneous nucleation).

– In unperturbed weak large scale updraughts subvisible cirrus clouds can exist in two different qualitative states, reaching either an equilibrium point in the long term behaviour or experiencing oscillation behaviour in a limit cycle scenario. The state depends on external parameters as large-scale updraught and temperature, respectively.

– The cloud particle properties in the long-term behaviour are very similar for both states. Therefore, we cannot decide from values of microphysical properties in a certain range in which state the cloud might be. Even if we had more measurements, we probably would not be able to decide between the two states just using the Eulerian measurements without a Lagrangian point of view.

– The derived bifurcation diagram may be interpreted as a minimal model for subvisible cirrus clouds, i.e. a damped oscillator, which changes its eigenvalues depending to environmental parameters $w$ and $T$, respectively, in a Hopf bifurcation.

We might derive a minimal model for SVCs from the bifurcation diagram in the following way. If we assume that SVCs are well approximated by their attractors, we could express cloud variables and relative humidity by a simple damped harmonic oscillator of the form

$$\ddot{x} + \kappa \dot{x} + \omega = 0, \tag{31}$$

with $x \in \{N_c, q_c, RH_i\}$ and parameters $\kappa = \kappa(w,T)$ and $\omega = \omega(w,T)$, respectively. $\kappa$ describes damping, whereas $\omega$ represents oscillation frequency. $\kappa, \omega$ can be determined using eigenvalues $\lambda_i$ for damping and oscillations in the point attractor case ($\kappa \neq 0$). For the limit cycle case ($\kappa = 0$), periods as obtained from the Poincaré section (see figure 11) can be used for describing $\omega$. Such a
minimal model could be used for representing SVCs in large-scale models and can be seen as a prototype for new generation cloud parameterisations. These models describe the structure of clouds in terms of cloud variables and environmental conditions. They could be used for describing such structures embedded into a coarse grid model. However, further research in this direction is necessary in order to proceed from pure model prototypes to useful cloud parameterisations.

Finally, we can state that we could develop a meaningful simple model for describing the main features of subvisible cirrus clouds. Former investigations using box-models indicated that there might be different regimes in the behaviour of the clouds for longer simulation times. For instance, in studies by Kay et al. (2006) and Spichtinger and Cziczo (2010) oscillatory behaviours as well as attractors could be seen. However, only a detailed mathematical analysis could show that there is a
bifurcation in the long-term behaviour and that it depends mostly on environmental parameters as updraught velocity and temperature. This analysis was only possible since we developed an analyt-

ical model, which is close enough to complex models but is also simple enough for mathematical analysis.

The observed Hopf bifurcation as a transition between two different states shows that clouds might exhibit inherent structures, which are crucially determined by the microphysical cloud processes themselves in addition to environmental conditions. Similar structure formation was already seen in analytical cloud models for liquid and mixed-phase clouds as developed by Wacker (1992, 1995, 2006) or Hauf (1993). Investigation and analysis of the microphysical processes in terms of sets of ordinary differential equations are a first but urgently necessary step in order to investigate structure formation inside clouds. Once we understand the possible structures in clouds as determined by microphysics, we can continue to further investigate structure formation as driven by spatial diffusion processes, mixing and others, leading to spatial structures of clouds. A first possible approach might be to investigate equations with additional spatial diffusion terms regarding possible Turing instabilities (Turing, 1952). However, further research in this direction is necessary in order to investigate structure formation of ice clouds.

## Appendix A:  Derivation of model equations

Splitting up the velocity as explained in section 2.1, $\boldsymbol{u}(m,\boldsymbol{x},t) = \boldsymbol{v}(\boldsymbol{x},t) + \boldsymbol{v}'(m,\boldsymbol{x},t)$, we adapt equation (1) accordingly:

$$\frac{\partial(\rho f)}{\partial t} + \nabla_x \cdot (\rho \boldsymbol{v} f) + \nabla_x \cdot (\rho \boldsymbol{v}' f) + \frac{\partial(\rho g f)}{\partial m} = \rho h. \tag{A1}$$

To derive equations for the evolution of moments, we multiply equation (A1) by $m^k$ and integrate by parts, using $f(0,\boldsymbol{x},t) = 0$ and $f(m,\boldsymbol{x},t) \to 0$ for $m \to \infty$, which are physically reasonable assumptions. This yields the following equation:

$$\frac{\partial(\rho \mu_k)}{\partial t} + \nabla_x \cdot (\rho \boldsymbol{v} \mu_k) + \nabla_x \cdot \left( \int_0^\infty m^k \rho \boldsymbol{v}' f \, \mathrm{d}m \right) =$$

$$k \int_0^\infty m^{k-1} \rho g f \, \mathrm{d}m + \int_0^\infty m^k \rho h \, \mathrm{d}m, \qquad k \in \mathbb{R}. \tag{A2}$$

We allow generalised moments $\mu_k$ with $k \in \mathbb{R}_{\geq 0}$, which occur naturally from cloud physics parameterisations. Formally, the unit of the $k$-th moment is $\mathrm{kg}^k \mathrm{kg}^{-1}$. For simplicity, we assume the mass distribution to be horizontally homogeneous, i.e. $f = f(m,z)$.

Using $\boldsymbol{v}' = (0,0,-v_t(m))$, and with the help of the continuity equation,

$$\frac{\partial \rho}{\partial t} + \nabla_x \cdot (\rho \boldsymbol{v}) = 0, \tag{A3}$$

the moment equation (A2) is rearranged to obtain equation (3).

## Appendix B: Details of parameterisations

### Nucleation

Homogeneous nucleation, i.e. the transformation of a solution droplet to an ice crystal, can be seen as a stochastic process. The transition rate $\omega$ for the transformation of a solution droplet of volume $V$ can be expressed using a volume nucleation rate $J$, i.e. $\omega = V \cdot J$. The probability $P(t)$ for the nucleation process of droplets of volume $V$ fulfil-ls the following differential equation:

$$\frac{\mathrm{d}P}{\mathrm{d}t} = -\omega P(t). \tag{B1}$$

For further details of the general derivation we refer to Koop et al. (1997). Equation (B1) can be generalised for size distributions of solution droplets, leading to the formulation of equation (7).

Koop et al. (2000) provide a parameterisation for the volume nucleation rate $J$ as a function of $\Delta a_w := a_w - a_w^i$ (Koop et al., 2000, Table 1, eq. 7). Here $a_w$ is the water activity of the solution and $a_w^i$ is the water activity of the solution in equilibrium with ice. Note, that the freezing characteristics of the droplets do not depend on the chemical composition. By definition the water activity is the ratio $p_{sol}/p_{liq}$ of the vapour pressure over a solution, $p_{sol}$, and pure liquid water, $p_{liq}$. Neglecting the Kelvin effect and assuming that the solution droplets are in equilibrium with the environment ($p_v = p_{sol}$), the water activity is proportional to the water activity in equilibrium with ice, which is the ratio of the water vapour pressure over ice and pure liquid water:

$$a_w = \frac{p_{sol}}{p_{liq}} = \frac{p_v}{p_{liq}} = \frac{RH_i}{100\%} \frac{p_{si}}{p_{liq}} = \frac{RH_i}{100\%} a_w^i. \tag{B2}$$

Both $p_{si}$ and $p_{liq}$, only depend on temperature and are parameterised according to Murphy and Koop (2005, eq. 7 and 10, respectively). Hence, $\Delta a_w$ is a function of $RH_i$ and $T$, as given by

$$\Delta a_w(T, RH_i) = \left(\frac{RH_i}{100\%} - 1\right) a_w^i(T) = \left(\frac{RH_i}{100\%} - 1\right) \frac{p_{si}}{p_{liq}}. \tag{B3}$$

Therefore $J$ is also a function of $RH_i$ and $T$. The logarithm of the nucleation rate is parameterised by a third order polynomial in $\Delta a_w$ (Koop et al., 2000, table1, eq. 7):

$$\log_{10} J(T, RH_i) = -906.7 + 8502 \, \Delta a_w - 26924(\Delta a_w)^2 + 29180(\Delta a_w)^3. \tag{B4}$$

### Diffusional growth

The "advection velocity" $g$ in the mass space is given by the growth equation for a single ice crystal; this equation has the following form (see, e.g., Stephens, 1983):

$$g(m) = \frac{\mathrm{d}m}{\mathrm{d}t} = 4\pi C D_v^* \rho(q_v - q_{v,si}) f_v. \tag{B5}$$

Here, $q_{v,si} = \varepsilon \, p_{si}(T)/p$ denotes the saturation mixing ratio, the shape of the ice crystal is accounted for by the capacity $C$ (assuming the electrostatic analogy, see e.g. McDonald, 1963; Jeffreys, 1918), $D_v^*$ is the full diffusion constant including the kinetic correction for small particles (Lamb and Verlinde, 2011) and $f_v$ denotes the ventilation coefficient.

In this study we make use of the following simplifications:

1. Latent heat release at the crystal surface is neglected and the temperature of the ice particles is assumed to be equal to temperature of ambient air.

2. We neglect kinetic corrections, since we are mostly interested in growth of larger crystals. Kinetic corrections are usually important for ice crystal growth in regimes with high concentrations of small crystals. For SVCs we can assume small concentrations, thus crystals will grow fast to sizes larger than $\sim 10\,\mu m$. Thus, we can assume

$$D_v^* \approx D_v = D_0 \left(\frac{T}{T_0}\right)^\alpha \left(\frac{p_0}{p}\right),$$ (B6)

with $D_0 = 2.11 \cdot 10^{-5}\,\mathrm{m^2 s^{-1}}$, $T_0 = 273.15\,\mathrm{K}$, $p_0 = 101325\,\mathrm{Pa}$, $\alpha = 1.94$ (e.g. Pruppacher and Klett, 1997).

3. We neglect correction of ventilation, setting $f_v = 1$. Ventilation correction is only relevant for very large crystals, so this is a reasonable assumption, since in SVCs ice crystals are usually smaller than $\sim 200\,\mu m$.

4. The shape of ice crystals is assumed to be prolate spheroids with length $L$ and an eccentricity $\varepsilon'$, which leads to the following expression (McDonald, 1963):

$$C = L \frac{\varepsilon'}{\log\left(\frac{1+\varepsilon'}{1-\varepsilon'}\right)}.$$ (B7)

For the mass-length relation we assume a simple power law $L(m) = C_i m^{\alpha_i}$ using $C_i = 1.02\,\mathrm{m}$, $\alpha_i = 0.4$. This power law mostly represents the columnar shape of ice crystals, which is assumed for crystals with sizes $L > 10\,\mu m$. The power law was fitted to a more complex description in Spichtinger and Gierens (2009), where a transition between droxtals and columns is formulated and used.

The fraction in equation (B7) only depends weakly on the crystal mass and can be approximated by a constant mean value of $1/3$. This yields

$$C = \frac{1}{3} C_i m^{\alpha_i}.$$ (B8)

With these assumptions, equation (B5) can be approximated as follows:

$$g(m) \approx \frac{4}{3}\pi C_i D_v m^{\alpha_i} \rho(q_v - q_{v,si}) = \frac{4}{3}\pi C_i D_v m^{\alpha_i} \rho q_{v,si}(S_i - 1),$$ (B9)

leading to equation (11).

**Sedimentation**

The description of sedimentation is based on the concept of mass and number weighted terminal velocities defined by Spichtinger and Gierens (2009). An expression for the sedimentation flux (i.e.

the integral in the sedimentation term in equation (3)), can be found by applying the mean value theorem. Consider a mean velocity, $\bar{v}_k$, such that

$$\int_0^\infty v_t(m)\rho m^k f(m)\,\mathrm{d}m = \bar{v}_k \int_0^\infty \rho m^k f(m)\,\mathrm{d}m = \rho\bar{v}_k\mu_k. \tag{B10}$$

There exists a corresponding velocity for each moment of the distribution $f(m)$. For the double moment scheme, the number weighted terminal velocity (for the number flux), $\bar{v}_0 = \bar{v}_n$ $(k = 0)$ and the mass weighted terminal velocity (for the mass flux), $\bar{v}_1 = \bar{v}_q$ $(k = 1)$, are relevant. For the calculation of the weighted velocities, we use a special representation of $v_t(m)$.

The dependency of the fall speeds of individual ice crystals on the crystal mass is approximated
by a simple power law $v_t(m) = \gamma m^\delta \mathrm{corr}(T, p)$, including a temperature and pressure dependent density correction factor,

$$\mathrm{corr}(T, p) = \left(\frac{p}{p_{00}}\right)^{a_i}\left(\frac{T}{T_{00}}\right)^{a_2}, \tag{B11}$$

with $T_{00} = 233\,\mathrm{K}$, $p_{00} = 300\,\mathrm{hPa}$, $a_1 = -0.178$, $a_2 = -0.397$. The coefficients $\gamma = 63292.36\,\mathrm{ms^{-1}kg^{-\delta}}$ and $\delta = 0.57$ are assumed to be constant over the entire range of $m$, as opposed to piece wise con-
810 stant values in Spichtinger and Gierens (2009). This approximation is justified since we assume ice crystals of sizes in the range between $\sim 10\,\mathrm{\mu m}$ and $\sim 200\,\mathrm{\mu m}$ for SVCs. The weighted velocities for number and mass flux, respectively, have the following form:

$$\bar{v}_0 = \bar{v}_n = \gamma\frac{\mu_\delta}{\mu_0}\cdot\mathrm{corr}(T, p), \ \ \bar{v}_1 = \bar{v}_q = \gamma\frac{\mu_{\delta+1}}{\mu_1}\cdot\mathrm{corr}(T, p). \tag{B12}$$

**Coefficients**

For simplification of the representation of the main system, we introduced coefficients in equations (25). In the following the coefficients are provided.

$$a = \frac{4}{3}\pi N_a\mu_{3,a}[r] \tag{B13a}$$

$$b = \frac{\gamma}{\Delta z}c(T, p)r_0^{\frac{\delta(\delta-1)}{2}} \tag{B13b}$$

$$c = \frac{\gamma}{\Delta z}c(T, p)r_0^{\frac{\delta(\delta+1)}{2}} \tag{B13c}$$

$$d = \frac{4}{3}\pi C_i\varepsilon\rho D_v\frac{p_{si}(T)}{p}r_0^{\frac{\alpha_i(\alpha_i-1)}{2}}\frac{1}{100\%} \tag{B13d}$$

$$e = g\frac{M_{air}}{RT}\left(\frac{L_{ice}}{c_pT} - 1\right) \tag{B13e}$$

$$f = \frac{4}{3}\pi C_i\varepsilon\rho D_v r_0^{\frac{\alpha_i(\alpha_i-1)}{2}} \tag{B13f}$$

## Appendix C: Derivation of eq. (28)

For deriving equation (28) we start with the slightly simplified systems of equations:

$$a \cdot J(RH_i, T) - b \cdot N_c^{1-\delta} q_c^{\delta} = 0 \tag{C1a}$$

$$-c \cdot N_c^{-\delta} q_c^{\delta-1} + d \cdot (RH_i - 100\%) N_c^{1-\alpha_i} q_c^{\alpha_i} = 0 \tag{C1b}$$

$$e \cdot w \cdot RH_i - f \cdot (RH_i - 100\%) N_c^{1-\alpha_i} q_c^{\alpha_i} = 0 \tag{C1c}$$

We convert equation (C1b) into the following form, using the mean mass $\overline{m} = q_c/N_c$ for cloudy states ($N_c \neq 0$):

$$-c \cdot \overline{m}^{\delta} + d \cdot (RH_i - 100\%) \overline{m}^{\alpha_i - 1} = 0. \tag{C2}$$

From this equation we obtain a representation for the mean mass:

$$\overline{m} = \left( \frac{d}{c} (RH_i - 100\%) \right)^{\frac{1}{\delta + 1 - \alpha_i}}. \tag{C3}$$

In a similar way, we can rearrange equation (C1a) for a representation of $N_c$:

$$N_c = \frac{a \cdot J(RH_i, T)}{b} \cdot \overline{m}^{-\delta}. \tag{C4}$$

Using equations (C3) and (C4) in equation (C1c) we obtain equation (28). The roots w.r.t. $RH_i$ of this equation are calculated using Newton's method.

## Appendix D: Example for a Poincaré section

In figure 16 we present an example of a Poincaré section, as used for the determination of the limit cycle. The plane, $\Sigma$, is such that $RH_i$ is constant on $\Sigma$ and $x_0$ is in $\Sigma$. Two different scenarios are represented here. First, we use a point close to the unstable focus point as initial condition for the numerical integration (indicated by red cross). The red dots indicate the section of the trajectory with the transversal plane $\Sigma$. The red dots converge fast to two accumulation points, which determine approximately the section of the limit cycle with the plane $\Sigma$. If we start "outside" of the limit cycle, the section points (indicated by blue dots) again converge fast to the same two accumulation points.

*Acknowledgements.* We thank M. Baumgartner, M. C. Papke, L. Grüne and R. Klein for fruitful discussions. This study was prepared with support by the German "Bundesministerium für Bildung und Forschung (BMBF)" within the HD(CP)[2] initiative, project S4 (01LK1216A) and with support by the German Research Foundation (DFG) within the Transregional Collaborative research Center TRR165 "waves to weather", project A2.

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

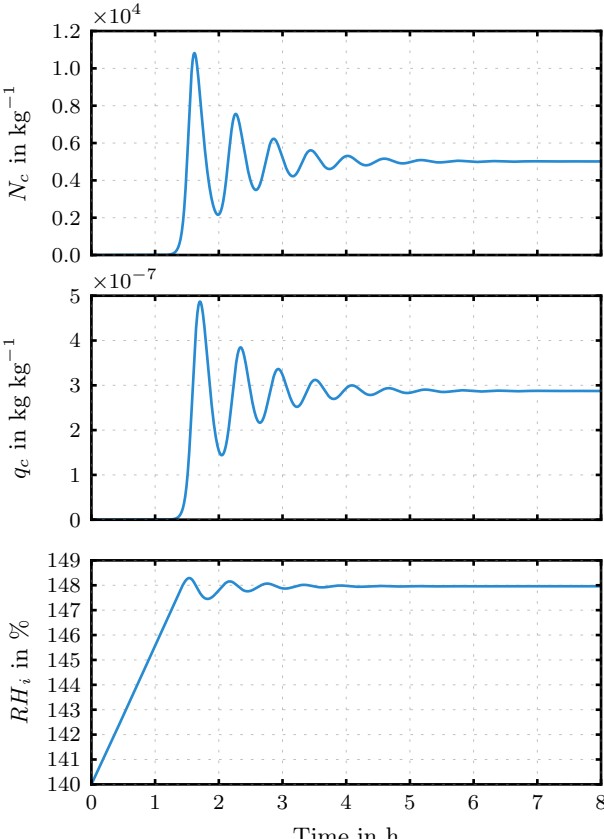

**Figure 1.** A scenario in state 1 (point attractor regime, damped) at $w = 0.01\,\mathrm{m\,s^{-1}}$ and $T = 220\,\mathrm{K}$. The continuous nucleation as well as similar time scales of nucleation, growth and sedimentation lead to a damped oscillation with an equilibrium state for $t > 7$ h. In phase space, the attractor property is more obvious (see figure 5).

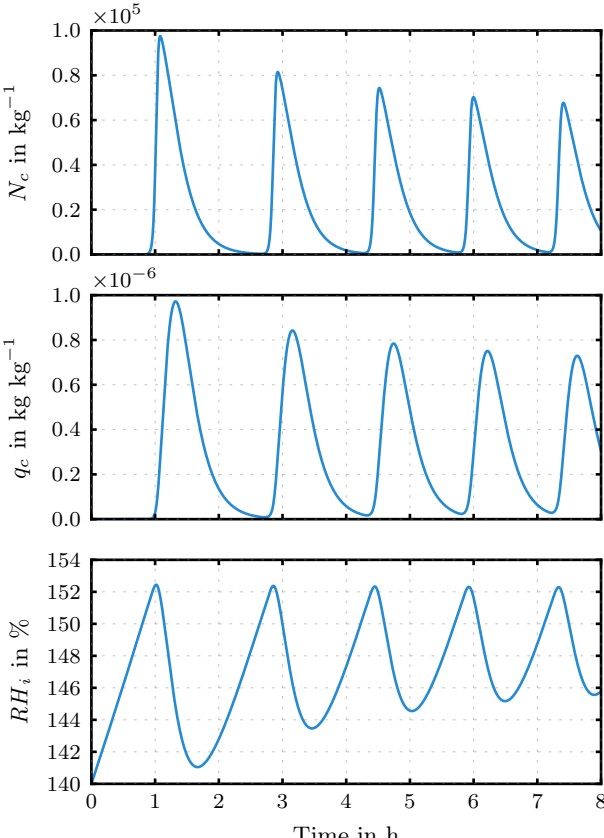

**Figure 2.** A scenario in state 2 (limit cycle regime) is shown at $w = 0.02\,\mathrm{m\,s^{-1}}$ and $T = 210\,\mathrm{K}$. Nucleation events occur as pulses, thus an undamped oscillation evolves, which describes a limit cycle in phase space (see figure 6).

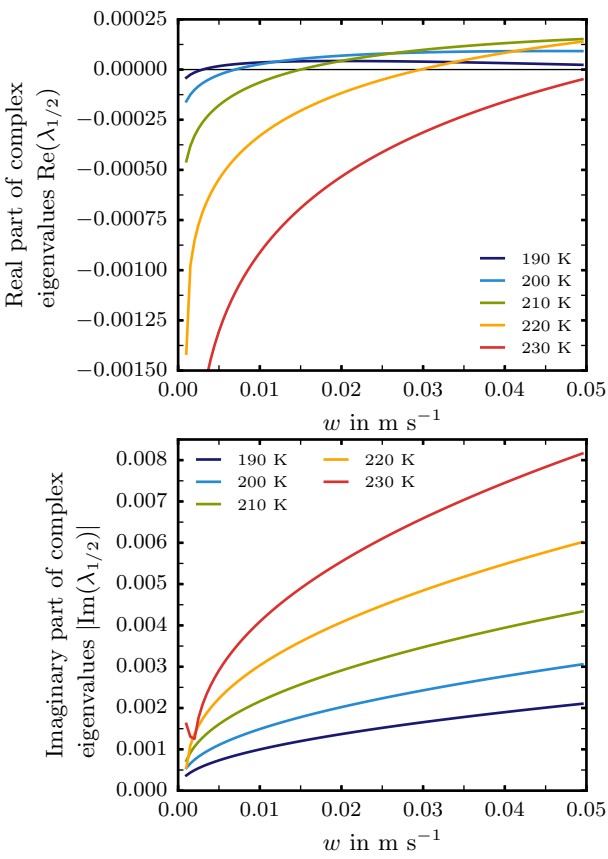

**Figure 3.** Real (upper panel) and imaginary part (lower panel) of the complex eigenvalues $\lambda_{1,2}$ of the Jacobian $\mathbf{DF}|_{\boldsymbol{x}_0}$ at the equilibrium point $\boldsymbol{x}_0$.

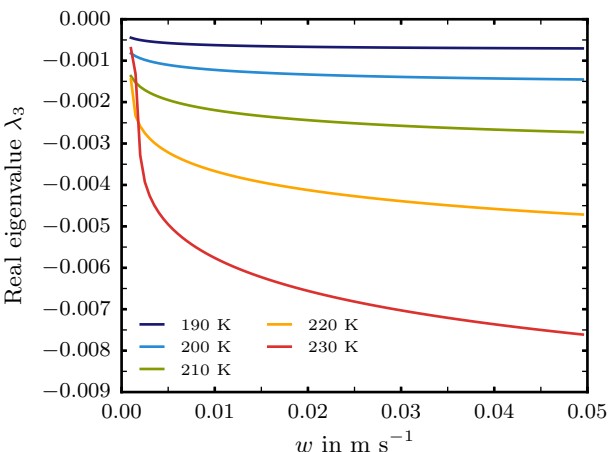

**Figure 4.** Real eigenvalue $\lambda_3$ of the Jacobian $\mathbf{DF}|_{\boldsymbol{x}_0}$ at the equilibrium point $\boldsymbol{x}_0$.

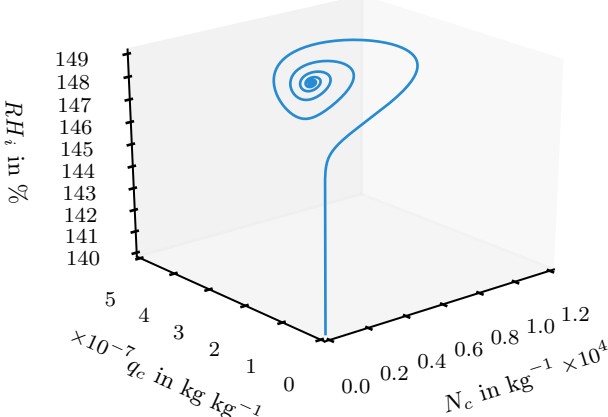

**Figure 5.** Positive point attractor for state 1 at $T = 220\,\mathrm{K}$, $w = 0.01\,\mathrm{m\,s^{-1}}$: orbit in phase space approaching the equilibrium point.

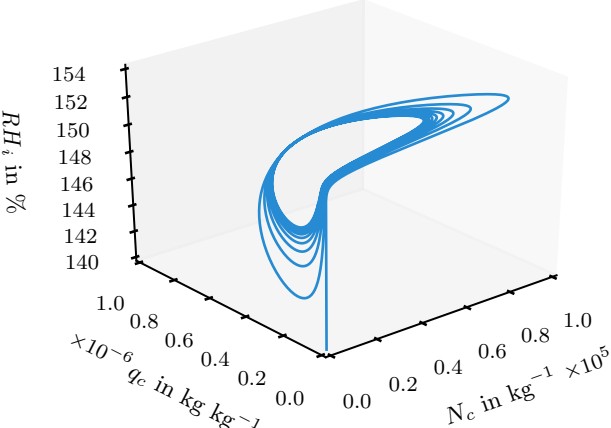

**Figure 6.** Limit cycle for state 2: orbit in phase space at $T = 210\,\mathrm{K}$, $w = 0.02\,\mathrm{m\,s^{-1}}$. Note that the solution starts "outside" of the limit cycle and approaches the limit cycle attractor asymptotically.

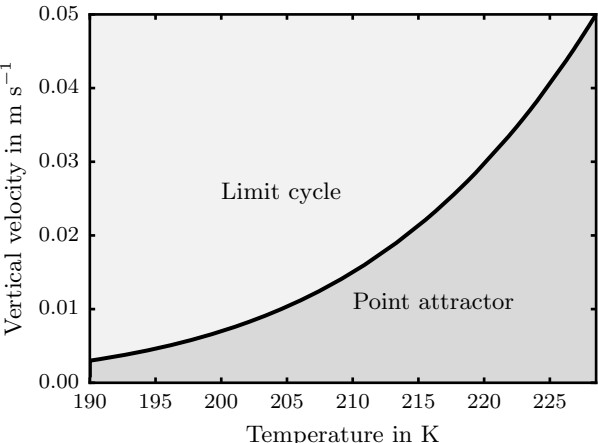

**Figure 7.** Bifurcation diagram for "positive point attractor" (state 1) and "limit cycle" (state 2) regimes in the $w$-$T$-space. The thick line indicates the location of the Hopf bifurcation.

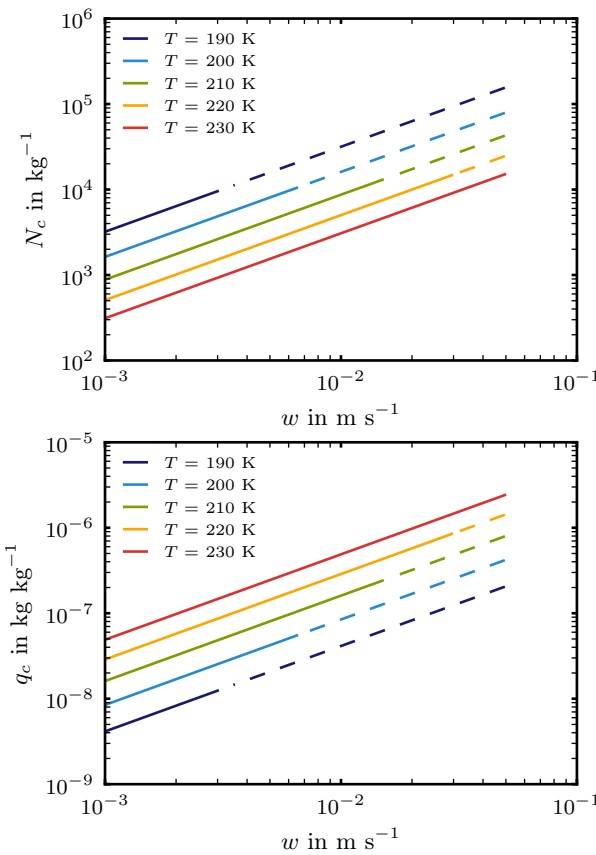

**Figure 8.** Ice particle number concentration $N_c$ (upper panel) and ice particle mass concentration $q_c$ (lower panel) at the critical point as a function of vertical velocity for different temperatures. Solid lines indicate parameter combinations $(w, T)$ in the point attractor regime (state 1), dashed lines represent the limit cycle regime (state 2).

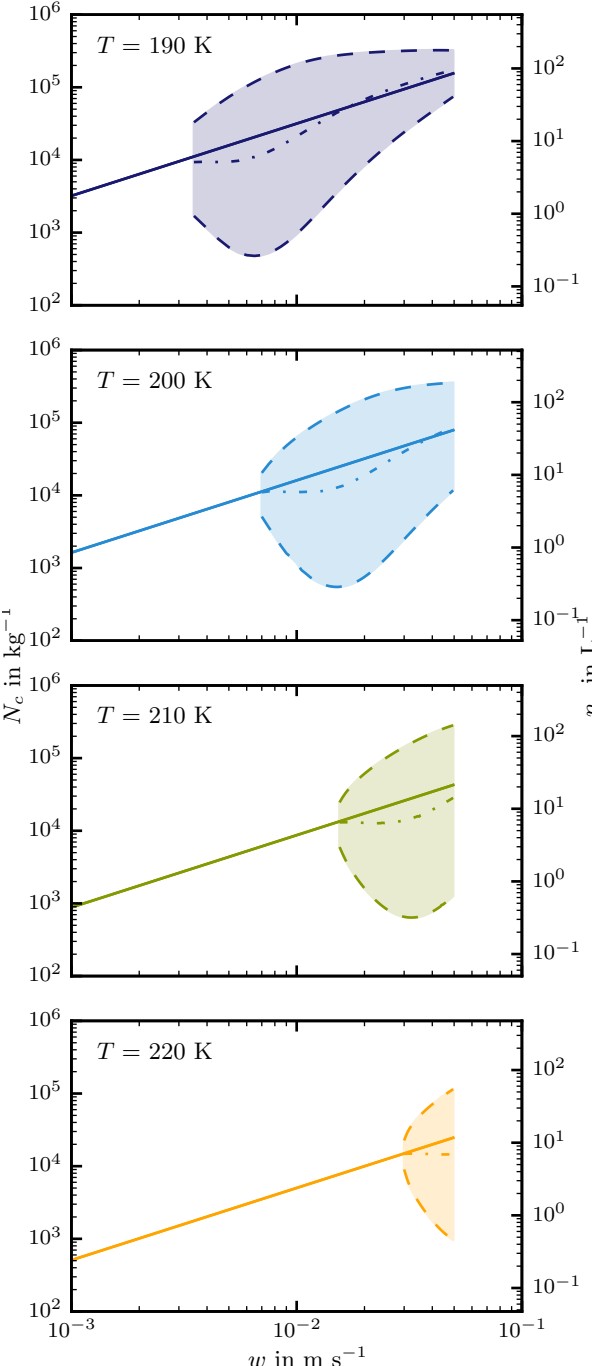

**Figure 9.** Ice crystal number concentrations for different temperature scenarios ($T = 190, 200, 210, 220\,\mathrm{K}$). The solid line represents values at the critical point $\boldsymbol{x}_0$ (stable or unstable focus). For the limit cycle regime the range of ice crystal number concentrations is indicated by the shaded area bounded by minimum and maximum values for the updraught range $0.001 \leq w \leq 0.05\,\mathrm{m\,s}^{-1}$; the median is indicated by the dot-dashed line.

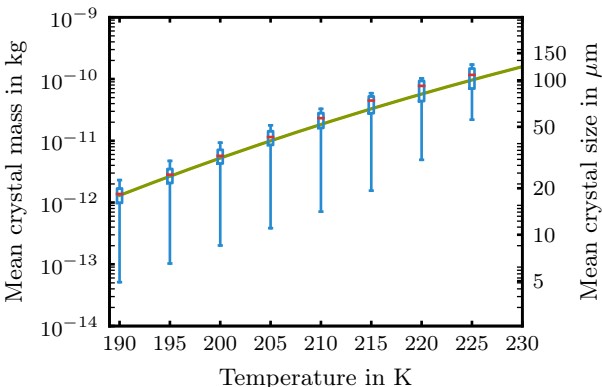

**Figure 10.** Mean ice crystal mass $\overline{m}$ as a function of temperature. For the critical point $x_0$, values of $\overline{m}$ depends only slightly on the vertical velocity, the curve covers the area that corresponds to vertical velocities $0.001\,\mathrm{m\,s^{-1}} \le w \le 0.05\,\mathrm{m\,s^{-1}}$. Additionally, box and whiskers plots indicate median, 25%/ 75% percentiles, and minimum/maximum values, respectively, for the limit cycle regime.

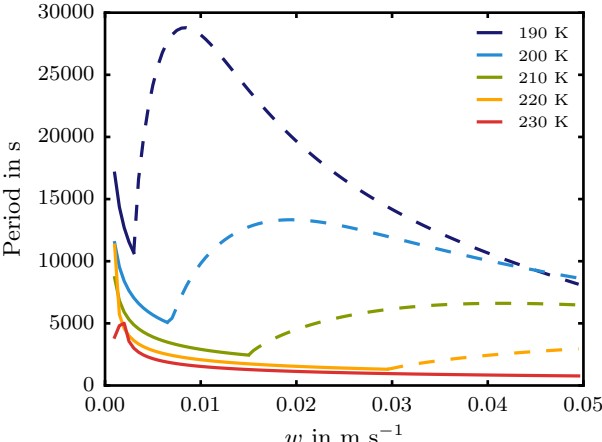

**Figure 11.** Oscillation periods for the point attractor regime at $x_0$ (solid lines), and for the limit cycle regime (dashed lines).

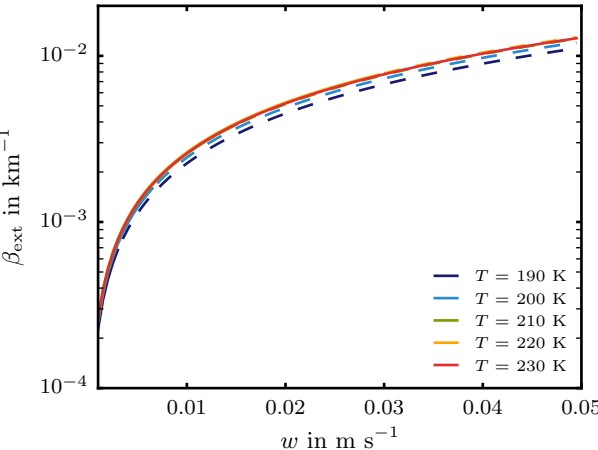

**Figure 12.** Extinction coefficient at $x_0$ for different temperatures in point attractor state 1 (solid lines) and limit cycle state 2 (dashed lines).

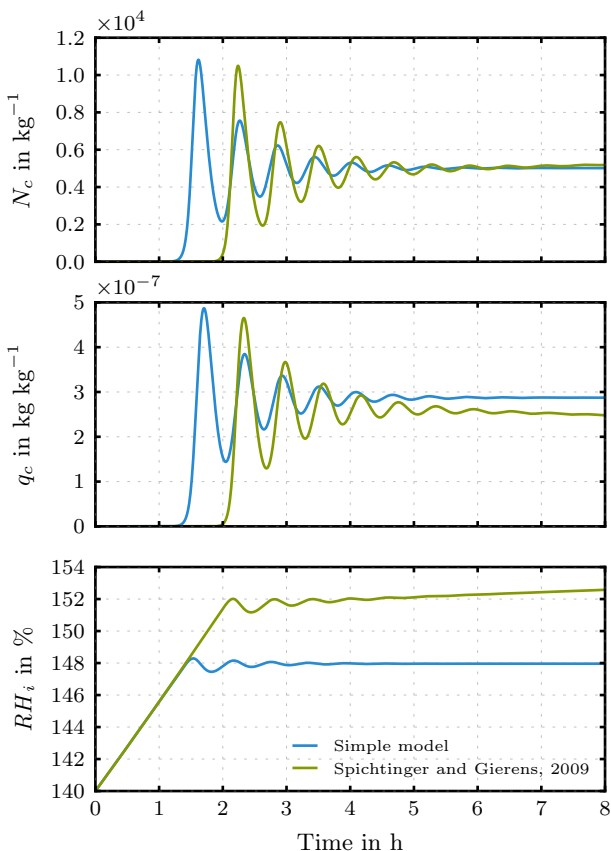

**Figure 13.** Point attractor case (state 1): Comparison between simple box model and the complex model by Spichtinger and Gierens (2009). Updraught $w = 0.01 \mathrm{\,m\,s^{-1}}$, temperature in the simple model and start temperature of the complex model is $T = 220\,\mathrm{K}$.

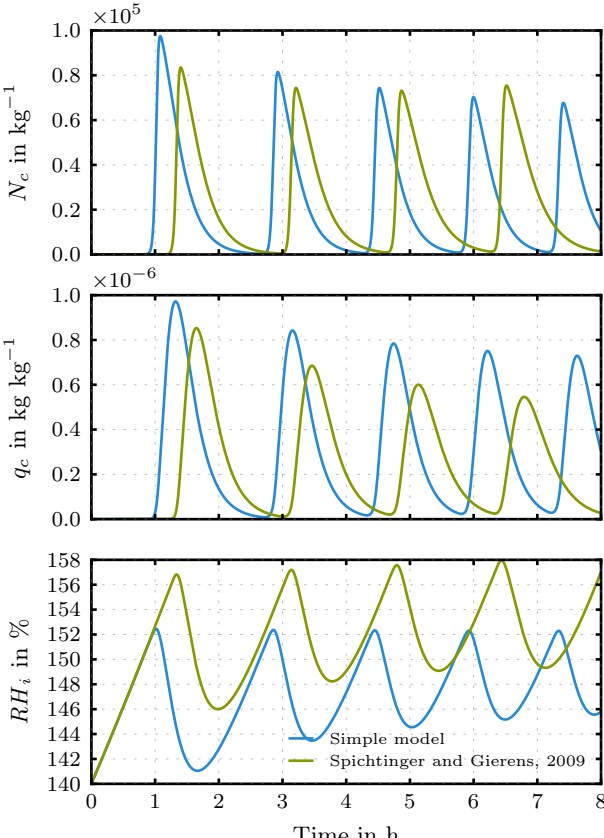

**Figure 14.** Limit cycle case (state 2): Comparison between simple box model and the complex model by Spichtinger and Gierens (2009). Updraught $w = 0.02 \text{ m s}^{-1}$, temperature in the simple model and start temperature of the complex model is $T = 210 \text{ K}$.

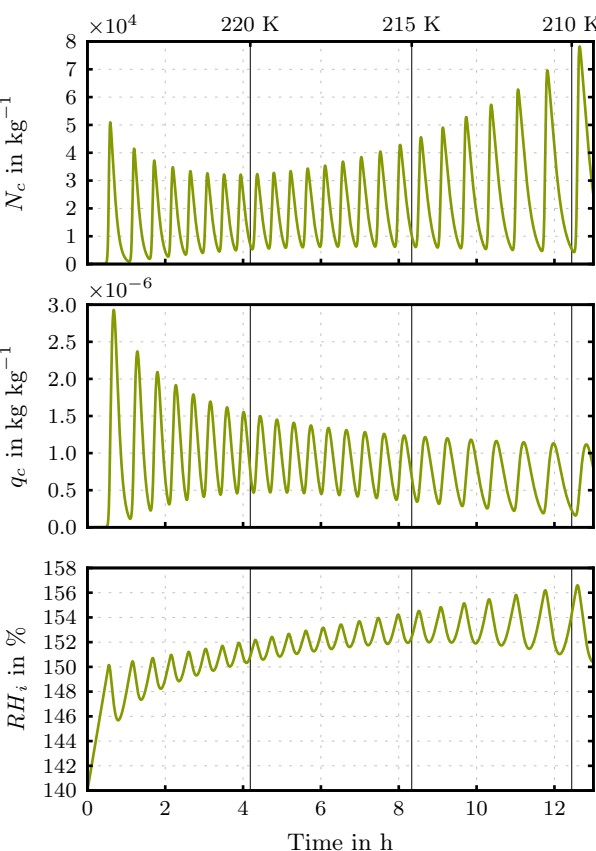

**Figure 15.** Transition between attractor regime (state 1) and limit cycle regime (state 2): Simulation with the complex model by Spichtinger and Gierens (2009) for $w = 0.035 \ \mathrm{m\,s^{-1}}$ and start temperature: $T = 225\,\mathrm{K}$. During the first two hours of the simulation, the attractor characteristics can be clearly seen. After reaching temperatures of about $T \sim 220\,\mathrm{K}$, the regime changes from state 1 (point attractor) to state 2 (limit cycle), see also phase diagram in fig. 7. After this transition, the amplitudes of number concentrations and relative humidity w.r.t. ice increase and at the end of the simulation also a shift in the oscillation period can be seen. Increase in amplitude and shift in oscillation period are due to changes of the limit cycle properties for decreasing temperature (see, e.g., figure 11)

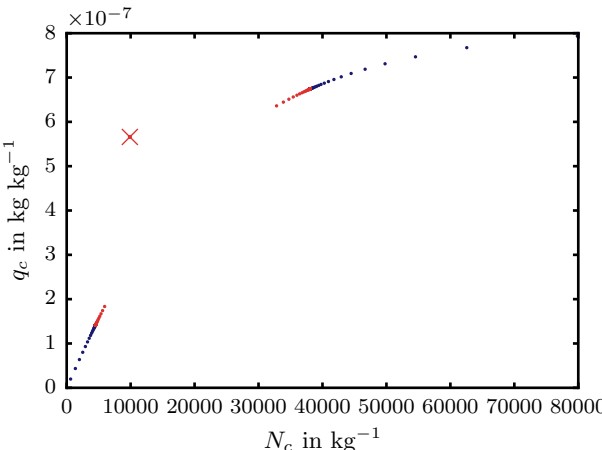

**Figure 16.** Example of a Poincaré section in the limit cycle regime. Blue dots indicate intersection points of the trajectory with $\Sigma$ when starting "outside" the cycle, red dots indicate intersection points when starting near the (unstable) equilibrium point $\boldsymbol{x}_0$ (red cross).