# Peer review of "Subvisible cirrus clouds – a dynamical system approach"

_Nonlinear Processes in Geophysics, 2015_

## Referee Comment (RC1) · Anonymous Referee #1 · 14 Mar 2016

Review of 'Subvisible cirrus clouds – a dynamical system approach' by E. J. Spreitzer, M. P. Marschalik and P. Spichtinger

The article introduces a new model to describe the evolution of subvisible cirrus clouds based on first principles, interpreting the results using concepts of dynamical systems theory. While the work is interesting and important for advancing the modelling of clouds in numerical atmospheric models, I feel that some aspects need to be corrected before it can be published in NPG. In particular, I think that the use of the dry adiabatic approximation as an assumption of the air parcel's vertical motion needs stronger justification. Also, there are several sections comprising detailed descriptions of general dynamical systems theory that do not seem to be used in the investigation. Finally, equation (42) appears to be incorrect. A less crucial, yet important point, is the use of very informal language throughout the in the manuscript. I give more detailed comments in the following lines:

**Major comments**

L252-256 and L288: Given that the model describes phase changes it is difficult to see that the motion can be described as dry adiabatic. There will be significant latent heat exchange. The motion could be assumed moist adiabatic but not dry adiabatic.

L363-364: How long does it take to reach the limit cycle? And how can we be sure that it is a limit cycle and not a very slow approach to a different kind of attractor? Fig. 2 shows the amplitude of the oscillations slowly decreasing rather than going into tending to a motion with stable amplitude.

L377-380: Please double-check that (42) is correct. I am getting different exponents for $r_0$ and $RH_l$ so that the factors involving these variables would be:

$$r_0^{\frac{\alpha_i - \delta}{2}}$$

for $r_0$, and

$$RH_i(RH_i - 100\%)^{\frac{1}{\alpha_i - 1 - \delta}}$$

for $RH_i$.

If (42) is correct, please provide more details for its derivation. Also it is not clear how this equation was used to produce any results. Was Fig. 5 produced by solving (42)? If so, please make sure it corresponds to the correct version of the equation. Figure 6 and 7 might also be affected.

L384-406: Dynamical systems theory is invoked for the interpretation of the results from the numerical model. However, I feel that this interpretation is very descriptive and not very quantitative. For example, the eigenvalues are never shown to prove that indeed they are behaving as described by the theory. Please show the eigenvalues and how they change as the parameter space is explored.

L407-413: In this paragraph it is stated that the limit cycle was determined numerically and the construction of a Poincare map is described. But then, without showing any results, in the last sentence it is stated that this is of no further interest! I don't understand why to determine the limit cycle and why to describe so much about the procedure if no results will be presented and no discussion will be made. Please show this results or else delete the seemingly unnecessary paragraph.

**Minor comments**

L11: A limit cycle can be an attractor. It might be worth revising the terminology (i.e. attractor v limit cycle) or providing a definition for these two concepts in the main text.

L21: Separate 'up to'

L24: It is implicitly claimed that the net effect of liquid clouds on the total energy budget is already known. I don't think this is correct, judging by the amount of research into that area. Please provide some references to back up this implicit claim.

L28: The *low temperature regime* is mentioned here for the first time, and no link to the clouds' environment has been made previously. In order to complete the background about the environment in which cirrus clouds exist, please explicitly state the order of temperatures and pressures in which subvisible cirrus clouds can be found. This can be done after the very first sentence of the Introduction.

L41: The sentence starting with 'For these subvisible clouds…' is not clear as written. What would warm, the clouds or the environment?

L42: The sentence starting with 'Our knowledge…' is poorly linked to the rest of the paragraph. Perhaps start a new paragraph.

L50: Why would sedimentation of ice crystals be important and so why should it be included?

L53: Change 'short:' for 'hereafter'.

L63: Change 'systems' for 'system'.

L63: Delete 'respectively'.

L65: What is meant by 'more general'? With respect to what is the new approach 'more general'?

L67: Where is 'here'? Perhaps change to 'In this article'.

L72: Delete 'respectively'.

L79: Define $\nabla_x$.

L83: Clarify what inertial system is being consider. And if it is inertial, why does its velocity depend on time ($\boldsymbol{v} = \boldsymbol{v}(\boldsymbol{x}, t)$)?

L90: What is meant by '(even with this simplification)'? the only assumption that has been introduced is the writing of total velocity as u=v+v', but I cannot see how this simplified equation (3). Please clarify.

L94 and L101: Is it right to define k as a real number? What is the advantage of defining generalized moments?

L104: Change 'components' for 'component'.

L105: It is not the gravitational *acceleration*, but the gravitational *force* that is balanced by *drag*. *Drag* is more appropriate a term then *friction*.

L113: Please do not assume that every reader is familiar with the terminology. Please provide references where the *usual* ansatz is used.

L134: For clarity, state that 'log' is the symbol used for 'natural logarithm' in the paper.

L142: Change 'Similar as' to 'Similar to'.

L151: Change 'assuming' to 'Assuming'.

L155: The sentence 'thus we can treat the integral as a constant' is not clear. Is it that because J is independent of r it can be treated as a constant in the integral?

L159: What are the units of the standard deviation $\sigma_r$?

L158-161: Are the modal radius, geometric standard deviation and total number concentration somehow justified by observations? If so, please provide references. Explain otherwise.

L202: Change 'neglet' for 'neglect'.

L205: Change 'this' for 'which'.

L208: The explanation beginning with ',which was fitted' is not clear. Were the coefficients $C_i$ and $\alpha_i$ derived from Spichtinger and Gierens (2009) model?

L213: Change 'we end' to 'we end up'. However, consider rewriting as this is not formal language.

L216: Is the definition of saturation ratio correct? Shouldn't it be $S_i = q_v / q_{v,si}$?

L244: Change 'Lagranian' to 'Lagrangian' here and in every place in which this word appears.

L246: Please state explicitly the relationship between (31) and $d(\rho\phi)/dt$. Notice that they are not equal.

L260: Rewrite the sentence starting with 'Hence'. The way it is written is very informal.

L261: Notice that ms$^{-1}$ means 'per millisecond' while what is meant is 'meters per second'.

L262: 'meaning that after twelve hours'. I am not sure what the point of this argument is. By what process would a weakly air ascent by sustained for twelve hours in this context?

L268: What is the meaning of the comma in the expression for sedimentation flux?

L270-271: In what way would you obtain a hyperbolic term in the equations?

L292-294: The sentence beginning with 'Note that' should be moved to the earlier discussion on the assumptions in page 9 and 10.

L296: Do you mean equation (37) or (38)?

L299-300: Why does DEP$_{RH}$ appears in both sides of the equation and how was the last line of (39) derived?

L315-318: I am not sure what is gained with the sentence including (41). Perhaps remove it.

L337: What is meant by 'further increase'? Relative humidity was already decreasing. Do you mean that it increased again? Please explain and possibly rewrite.

L340: Can you give a quantified range for temperature and vertical velocity rather than saying 'At rather high temperatures and slow vertical velocities'? Looking at Fig 5 it looks like temperature does not need to be 'rather high'.

L353: See comment to L340.

L392: What does 'positive attractor' mean?

L422: Delete 'respectively'.

L422: How was the critical point found? Was (42) used? Please explain.

L439-441: The statement beginning with 'As indicated' is formally correct but it could be made more precise by indicating that the bifurcation point depend on w.

L455: How are the *mean sizes* calculated?

L462-464: Please provide more details. How is the Spichtinger–Gierens model more complex than the model presented here? In what sense is it more realistic?

L479-480: The sentence is written in informal language. Please rewrite.

L487: Please qualify the assertion that 'this holds for ALL cases of damped oscillations'. What is its observational, numerical or experimental evidence?

L502 and L516: What is mean by 'theoretical' and 'theoretically determined'? Do these terms refer to the simple model or to the more complex model? Why any of these models can be considered theoretical?

L537: What do the *mean ice crystal sizes* represent? Could they be interpreted as the width, length or perhaps radius of a crystal?

L607: The sentence beginning with 'we can proceed' is not clear. Please rewrite.

L609: 'Last but not least' is informal language.

L610: It is claimed that the model presented in the article could be a prototype for a new generation of cloud parametrizations. However, for this model to be useful it would need to provide estimates of the effects that subvisible cloud formation have on the environment. This aspect has not been addressed in this paper. It would be worth expanding the discussion on this topic.

Figure 1 and Figure 2: 'Here,…' is informal language. Please rewrite.

Figure 7. The caption says that the median is indicated by the dashed line, but there are three dashed lines in each panel.

Figure 9. Is ms$^{-1}$ the correct unit?

---

## Short Comment (SC1) · 14 Mar 2016

**Review**

**Nonlinear Processes in Geophysics, Discussion paper**
**TITLE: Subvisible cirrus clouds – a dynamical system approach.**
**AUTHOR(S): E.J. Spreitzer, M. P. Marschalik, and P. Spichtinger**

**Summary**

For the investigation of sub-visible cirrus clouds as formed by homogeneous freezing a simple analytical cloud model is developed from first principles. The presented results are interesting and in general relevant and important for improving the understanding the evolution of cirrus cloud. However, some clarifications and minor corrections need to be introduced before publication, improving the physical understanding of the presented results. Especially, the equations written down are in a form which is difficult to understand, covering or disguising the used first principles. Therefore, I can recommend the manuscript for publication in Nonlinear Processes in Geophysics after **revision**, taking into account the following major and minor review points:

**Major review points**

1. Chapter 2, page 3, line 78: What is the meaning of the term "Boltzmann-type way"? Equation (1) is a balance equation of a mass distribution in space-time plus a divergence-term in an additional internal phase space. The meaning of this last term in the left hand side of (1) is beyond the classical fluid dynamical setting and this needs more explanation.

2. Chapter 2.5. Why do you change from the Eulerian to the Lagrangian description? Can you explain ore motivate this step in more details? Moreover, the two terms in (31) cannot be seen as a Lagrangian conservation law. $(d(\rho\phi)/(dt) = \partial(\rho\phi)/(\partial t) + $ advection of $\rho\phi$ and not the divergence of the flux $(\rho \, {\bf v} \, \phi.)$. This should be made clear.

3. Chapter 2.6, page 11, line 303-314: Here, the 3D-system of the ordinary differential equations is written down explicitly. However, in this form the internal dependence and overall structure of the model equations are hard to understand. Can you write the equations in a more principle way? Which terms are linear or non-linear, which terms are conservative, and which are dissipative, leading to the attractor in phase space? Can you calculate e.g. the divergence in the phase space of the used variables.

**Minor review points**

1. Chapter 2.5 page 9, line 244: The names of the two physicists and mathematics Lagrange and Euler should be written correctly; like Lagrangian viewpoint or Eulerian time and so on. Please check the whole manuscript in this respect.

---

## Referee Comment (RC2) · U. Wacker (Referee) · 2 May 2016

*Review of*
**Subvisible cirrus clouds – a dynamical system approach**
*by E.J. Spreitzer et al.*

**Summary:**
The authors present a conceptual model to investigate evolution characteristics by the methods of Nonlinear Dynamics. I appreciate the research presented here and I will appreciate more publications on similar topics. Yet, before publication, I suggest some points of straightforward amendment, primarily a stronger recurrent theme, i.e. a nonlinear model from cloud physics investigated by the methods of nonlinear dynamics, and a careful inspection of the model equations and their derivation. The major comments are given below. I have also compiled a list of minor comments, including hints to wording and to errors in the presented eqs. For shortness, I send this list directly to the authors.

**Major comments**

1. The presented research is devoted to the investigation of a cloud physical problem in the context of the theory of dynamic systems. This should be the red line in the presentation. Some suggestions:
(a) Introduction: Reduce the discussion of SVCs, and put mor emphasis on the major point of the paper, i.e. a conceptual model using the ideas of Nonlinear Dynamics, which is mentioned only casually in l. 60-64 and in the last but one para of the conclusions. Other papers devoted to Nonlinear Dynamics and Cloud Physics are published by Graham Feingold, see e.g. Nonlin. Processes Geophys., 20, 1011-1021, 2013: 'A model of coupled oscillators applied to the aerosol-cloud-precipitation system' by G. Feingold and I. Koren.
(b) The central dynamic eqs.(40) can be written in a clearer way by introducing few coefficients, something like $dN_c/dt = aRH_i - bN_c^{1-\delta/\delta}q_c^\delta$ etc. Why is $RH_i$ not changed by nucleation? Why don't you use $q_v$ as prognostic variable instead of $RH_i$? Then one can see the condition of mass conservation immediately and get rid of a lot of calculations in Ch. 2. Another important advantage is that you can use the set of $(N_c, q_c, q_v)$-eqs. to characterize immediately the steady state by 2 conditions: (i) Nucleation and sedimentation of $N_c$ balance. (ii) Sedimentation of $q_c$ and increase of $q_v$ due ascent balance. A steady state is reached only for $RH_{i,steady} > 1$ and requires the parameter $w > 0$.
(c) The basic set of eqs. (40) comprises many terms. To discuss this set of ODEs in the frame of nonlinear dynamics, please discuss which of the terms represent the external sources and sinks, and which represent internal transformations.
(d) Discuss earlier Section 3.2 Mathematical Analysis as the heart of your method, and please give more explanations on Eq.(42). Once the set of prognostic equations is formulated, discuss the steady states and their (linear) stability (e.g. as in Section 3.2).

2. I have the impression that you use the wording 'critical points' instead of 'steady states'. Please correct and skip the sentence 'since the system...' in l. 371. A limit cycle is also an attractor, namely a periodic one. l. 392: What is a 'positive attractor'? A stable focus?

3. The authors assume dry adiabatic processes (Section 2.5) despite deposition. This requires clarification. Maybe, one can assume moist saturated adiabatic processes and use the value of the lapse rate at the prescribed temperature. For selected T = 233K or less, the lapse rate should be near 0.9 K/100m or even closer to the dry adiabatic one.

4. In Chapter 2, the basic equations are derived from first principles at length, while much has been derived earlier. The calculations can be reproduced at the utmost by lengthy derivations - which the reviewer did not do everywhere.
Please check whether the central equations and prerequisites of the final Eqs. (40) can be compiled in an appendix, accompanied by a considerable shortening of Chapter 2 and another contribution in favor of the recurrent theme. Also the reader would be happy to find a collection of relationships between the moments of the size distribution function and the used variables like $N_c$ and $q_c$, as well as a reduction of symbols to an absolute minimum number.

5. Chapter 2 carries inconsistencies in dimension. The chapter is devoted to the description of the spectral budget equation and the budget equation for the moments of the particle size distribution. This is well known from literature, also stated by the authors. However, the presentation does not agree with e.g., Beheng (2010). The authors start with a particle size distribution $f$ in dimension mass$^{-1}$ and a budget equation in terms of $\rho f$ with dimension volume$^{-1}$. How do you interpret $f dm$ and $\rho f dm$? The moments $\mu_k$ according to Eq.(4) have units mass$^k$. The zeroth and the first moments have dimensions of 1 and mass, resp., and cannot be identified as number concentration $N_c$ and mass concentration $q_c$ (l. 114) with units stated in l. 122. The dimension of $\mu_k$ is mass$^k$, not as in l. 102. $f(m)$ in eq.(8) would have the dimension mass$^{-2}$ with $N_c$ from l. 122. The outline of Beheng results in the correct properties.
Nucleation terms in (40a) and (17a) disagree, why?
l. 247: $\rho d\Phi/dt$.
Please clarify.

Ulrike Wacker

---

## Referee Comment (RC3) · Anonymous Referee #3 · 19 May 2016

Review of

**Subvisible cirrus clouds – a dynamical system approach**

E. J. Spreitzer, M. P. Marschalik and P. Spichtinger

The paper presents a new, straightforward theoretical approach to explain the formation mechanism and development of subvisible cirrus clouds (SVCs). Due to their climate feedback mechanisms and high frequency of occurance this is of importance, especially since this approach has the potential to be implemeted in large-scale models where a realistic representation of ice clouds is still problematic. Thus, the topic of the paper is of interest and the method is convincing. Its structure is clear and it is mostly well written. Altogether, the manuscript is a nice piece of work and I recommend it for publication in NPG.

Of course I have some suggestions that I think could improve the paper. The amount of comments lead to my rating of the paper as 'major revisions', though I have no major criticism. In complement to the other referees, who concentrated on the theoretical work, my comments focus on the SVCs itself. In this context,

a) I like to encourage the authors, reversely to the comment of RC2:

'*Reduce the discussion of SVCs, and put more emphasis on the major point of the paper, i.e. a conceptual model using the ideas of Nonlinear Dynamics, ...*'

to extend the discussion of SVCs. More detailed recommendations are given in the specific comments below.

b) I also like to know how realistic the modelsimulations are ? Specifically,

- will the two states develop in case of a previous heterogeneous ice nucleation event, which most probabaly occurs in the atmosphere?
- Also, will the two states develop in the presence of temperature fluctuations? Especially at low vertical velocities, temperature fluctuations can influence the development of cirrus clouds.

**Specific comments:**

1) Abstract: in its present form, I think the abstract might not highly attract the attention of potential readers. It could be written in a more clear way and give some more information about the results and impact of the study:

   (i) for example:

   '*We found two different states for the long-term behaviour of subvisible cirrus clouds, i.e. an attractor case and a limit cycle scenario. The transition between the states constitutes a Hopf bifurcation...*'

   Please say here that the attractor case is a damped and that the limit cycle scenario an undamped oscillation of the cirrus properties. Also, it would be more clear to say that the two states differ by the microphysical properties of the developing cirrus (attractor regime: stable cirrus at higher T with in general fewer and larger ice crystals, limit cycle regime: unstable cirrus at lower T with in general

more and smaller crystals) and are separated in the temparature – vertical velocity parameter space by a Hopf bifurcation.

(ii) it should be noted in the abstract that the study is valid for vertical velocities < 5 cm/s.

(iii) I also would recommend to include a sentence on the impact of the study to investigate 'structure formation' and also on the 'potential for large-scale models' – both points are now only mentioned at the end of the conclusions.

2) First part of Introduction: the word 'usually' appears quite often …

3) Introduction:
(i) the reference Hoose and Möhler (2012) is used for heterogeneous ice nucleation , but this process is already known much earlier. Isn't there a more basic reference?
(ii) line 36: you might also cite the articles of Immler et al. (2008), ACP, one study on mid-latitude and the other on tropical cirrus.

4) Page 9:    (i)  2.5 '*Additional settings*'  maybe better  '*Boundary conditions*' ?

(ii) I would recommend to provide a header   to  points 1. and  2., so that the reader has an immediate idea about the major settings.  E.g. :

1. *Langrangian coordinates*:  *Instead of an Eularian ….*

2. *Constant air parcel volume at cloud top and w < 5cm/s*:  *In our study, we will exclusively consider very low vertical velocities or vertical changes with limited amplitude.*
→  please also tell the reader here the reason to choose only very low vertical velocities – I think I know ;-) .

5) Page 10, first paragraph:   '*… meaning that after twelve hours the temperature difference would be about 8 K.*'   Would be good to note the magnitude of the change of Nc due to a density change caused by cooling by 8K is.

6) Page 11, first paragraph:   ODE  is not defined.

7) Page 12, end of second paragraph:
'*In fact, sedimentation is the key process, which leads to different states in the cloud evolution. The system exhibits two qualitatively distinct behaviours, depending on the parameter values of w and T.*'
Where is the link between the two sentences ?

8) Page 12, lines 340  and 353:      *State 1: …*   and    *State 2: …*
It would be helpful and more clear for the reader if the definitions are  more specific (here and throughtout the manuscript)

State 1 (attractor regime,  damped oscillations): …
State 2 (limit cycle regime,  undamped oscillations): …

9) Page 12, line 340:   '…, *the three competing microphysical processes …*'
Please define the processes, I guess you mean
' … , the three competing microphysical processes -nucleation, growth, sedimentation- *…*'

10) Page 12, lines 343-344: '*… resulting in a damped oscillation in all three variables,*

*finally asymptotically reaching an equilibrium ...*'
better
'... resulting in a damped oscillation in all three variables  Nc, qc, RHi, which finally
 asymptotically reaches an equilibrium ..'

11) Page 14, second paragraph:   Here I miss a more detailed discussion of Figure 5.

    What I see is that the higher the temperature, the higher w must be to switch the
    system from the damped attractor to the undamped limit cycle regime. That means
    that damped attractor cases are more frequent at higher temperatures and
    undamped limit cycle cases at lower  temperatures, yes ? Please discuss.

12) Page 14, line 424 and 425:
    '*correspond to state 1 (attractor regime, damped),...*'
    '*... for state 2 (limit cycle regime, undamped);*'

13) Paragraph page 14-15:   Also Figure 6 could be discussed in more detail.

    In this Figure it can be seen that most warmer cirrus are in the stable damped
    attractor regime, with few, large crystals. At colder temperatures there is a
    tendency to more variable cirrus with more, smaller crystals at higher w (limit cycle
    regime).

    A further consideration could  be if the cirrus in the limit cycle regime are still SVCs?
    Immler et al. (2008), ACP, provided relations between OD, IWP, Nc and Rc. Using
    these relations it should be possible to see which of the simulated cirrus are SVCs
    and which are thin cirrus – maybe only attractor regime cirrus are SVC?

    A last thought to Figure 6:  a third panel showing the mean size would be
    informative.

14) Page 15, lines 433 – 436:  'I*n the double logarithmic representation in figure 6, the
    number concentrations Nc(w) at x0 appear as straight lines. For different
    temperature regimes, there seems to be a constant shift between the curves Nc(w)
    (i.e. a constant factor c(T)), leading to parallel lines in the double logarithmic
    representation.*'

    Here you mean the attractor regime, yes ? Please state.

15) Page 17, first paragraph:   you use '*damped oscillations*' (line 500) together with
    '*limit cycle*' (line 504).  To ease the understanding of the manuscript, I recommend to
    always use 'attractor case (damped, state1)' together with  'limit cycle (undamped,
    state 2).

16) Page 19, line 568:
    '*The microphysical properties of the cloud in both states are similar ...*'
    I wouldn't say that they are  similar, see comment 13 !
    Same comment for lines 586-587.

**Comments to the Figures:**

Figure 1, caption: '*Here, a scenario in state 1 (damping) ...*'
        better     'Here, a scenario in state 1 (attractor regime, damping) ...'

Figure 2, caption: '*Positive attractor for state 2 (limit cycle) ...*'
        better     'Positive attractor for state 2 (limit cycle, no damping) ...'

Figure 5:  A suggestion

[Figure]

Figure 6:
- On the right axis, it would be very helpful if the values of   Nc in 1/L  and   qc in g/m3  which are noted in the text would be marked.
- Caption: '*Solid lines are for parameter combinations (w, T) in the attractor regime (state 1),  dashed lines are for the limit cycle regime (state 2).*'
  better    'Solid lines are for parameter combinations (w, T) in the attractor regime (damped, state 1), dashed lines are  medians of the the limit cycle regime (undamped, state 2), see also Fig. 7.'
- I would find it  informative to also see a panel showing mean sizes.

Figure 7, caption:  '*The solid line represents values at the critical point x0 of  the attractor regime.*', *yes ?*

Figure 8:
- Wouldn't it be worth to note in the text that the median state of the limit cycle regime is  nearly identical with the attractor regime (this is also seen in Figure 7 ....)?
- Caption:  '*For the critical point x0 of  the attractor regime ...* ', yes ?

---

## Author Comment (AC2) · 26 Sep 2016

For a detailed reply to the comments, we refer to the supplement of author's response.

---

## Author Comment (AC3) · 26 Sep 2016

For a detailed reply to the comments, we refer to the supplement of author's response.

---

## Author Comment (AC4) · 26 Sep 2016

For a detailed reply to the comments, we refer to the supplement of author's response.

---

## Author Response (AR1)

**Reply to referee's comments**

22 September 2016

**General comments**

We thank all reviewers for their comments, leading to an improvement of our manuscript. We collected some general comments, since some issues were addressed by all reviewers.

**Restructuring of the manuscript**

We restructured the manuscript in the following way:

- We generally revised the text to avoid misunderstanding, following many suggestions of the reviewers.
- We have shortened the derivation of the model and put the details into appendix B.
- We expanded the analysis part as well as the description of the analysis tools.
- We also expanded our comparison of the model results with observations in order to address the topic "subvisible cirrus clouds" adequately.

**Dry adiabatic motion, latent heat release, temperature/velocity regime**

If latent heat release was taken into account for the rate of  $RH_i$ , the temperature equation (18) would read

$$\frac{\mathrm{d}T}{\mathrm{d}t} = \frac{\mathrm{d}T}{\mathrm{d}z}\frac{\mathrm{d}z}{\mathrm{d}t} + \frac{\mathrm{d}T}{\mathrm{d}t}\bigg|_{\mathrm{latheat}} = -\frac{g}{cp}w + \frac{L_{ice}}{M_{air}c_p} \cdot \frac{\mathrm{d}q_c}{\mathrm{d}t}\bigg|_{\mathrm{growth}},\tag{1}$$

and the latent heat term would need to be included in the rate equation for  $RH_i$ :

,

$$\frac{\mathrm{d}RH_i}{\mathrm{d}t} = \underbrace{e \cdot w \cdot RH_i}_{\mathrm{adiabatic}} - \frac{\mathrm{d}q_c}{\mathrm{d}t} \bigg|_{\mathrm{growth}} \left( \underbrace{\frac{p}{\varepsilon p_{si}(T)} \cdot 100\%}_{\mathrm{growth}} - \underbrace{RH_i \frac{L_{ice}^2}{RT^2 M_{air} c_p}}_{\mathrm{latent heat}} \right).$$
(2)

The contribution to temperature change from latent heat is obviously only important, when there is substantial diffusional growth of ice particles, i.e. when  $\frac{dq_c}{dt}|_{\text{growth}}$  is considerably large. In that case however, the latent heat term directly competes with the growth term. The growth term is usually at least one order of magnitude larger than the latent heat term. Hence, whenever latent heat release comes into play for temperature changes, the rate of change of  $RH_i$  is dominated by the water vapour sink due to growth, anyway. Therefore, we omit latent heating in the  $RH_i$  equation and stick with the constant adiabatic forcing term. Including latent heating would produce an additional highly nonlinear term (it contains  $RH_i$  as a factor), which we would like to avoid.

In terms of lapse rates we can state that due to phase changes a "moist" (or better "ice") adiabatic lapse rate might be more appropriate. However, since we investigate cirrus clouds in the low temperature regime T < 235 K, the difference between moist adiabatic lapse rate and dry adiabatic lapse rate is less than 5% and decreases with decreasing temperature. Thus, we can approximate the temperature change by adiabatic lapse rate and omit the additional nonlinear term. This is also noted in the text.

We also describe the relevant regime for SVCs, i.e. low temperatures T < 235 K and slow vertical updraughts w < 0.05 m s-1 in the first part of the manuscript.

**Terminology**

We changed the term "(positive) attractor" to "(positive) point attractor" or "stable focus" for clarity. We use the term "critical point" according to Verhulst [1996], which is equivalent to "steady state" or "equilibrium point". We add some information that the limit cycle is a one-dimensional attractor.

**Response to Peter Nevier**

**Boltzmann equation** The evolution equation of the size distribution has a form that is similar to the Boltzmann equation (e.g. in gas dynamics). However, since we have no aggregation terms on the right hand side, we follow the suggestions and omit this name.

**Eulerian vs. Lagrangian description** Since we are interested in the time evolution of an air parcel, the change from Eulerian to Lagrangian description seems reasonable. We added more details for the transformation of the equations; especially, we describe the reformulation of the evolution equation in advective form, using mass conservation of air (appendix A).

**ODE system** We write the equation in a simpler way, using constants in order to represent the essential non-linear features of the equations. We added a paragraph for a qualitative description of the different terms (non-linear/linear, etc.) and calculated the divergence for determining the quality of the system (externally forced dissipative system).

**Spelling** We corrected all typos, especially the names of the physicists and mathematicians Euler and Lagrange.

**Response to reviewer #1**

**Major comments**

L252–256 See general comments.

**L363–364** As it is a limit cycle, it takes infinitely long for the orbit to reach it. However, in the numerical solution, the oscillation amplitude stays approximately constant after several periods. This is apparent from the results from the Poincaré sections. The sequence of intersection points reaches an accumulation point after  $\sim 50$  cycles. From then on, the points only fluctuate minimally about the asymptotic limit, with no long term trend.

**L377–380** We thank the reviewer for the careful recalculation. Equation (42) was indeed wrong. The error occurred when the equation was rearranged for better readability and transferred to  $\text{LAT}_{\text{E}}X$ -Code. In the actual calculations for the analysis, the correct equation was used. The correct equation is now provided (eq. 28) as well as a short derivation in appendix C. The equation was solved using Newton's method to obtain the critical point for different parameter values (w,T), and the corresponding Jacobians and eigenvalues thereof.

**L384–406** The dependence of the eigenvalues on w and T is shown in figures 3 and 4.

**L405–413** The paragraph describing the construction of the Poincaré section is now improved and accompanied by a more detailed explanation in appendix D. In fact, the numerical results are needed for the determination of the limit cycle. The periods of the limit cycle are shown in figure 11.

**Minor comments**

Comments regarding wording are not specifically addressed, since we rewrote the manuscript considering these suggestions.

**L11** See General comments.

**L24** We changed the text in this respect. In fact, the exact net effect of liquid clouds is not known yet. However, it is known that the albedo effect is usually dominant, thus liquid clouds usually have a cooling impact on Earth's energy budget. We cite now the latest IPCC report for this qualitative behaviour of warm clouds.

**L28** In the revised version, the low temperature regime is now mentioned in the first sentence of the introduction.

L41 We added more text, to make clear that they warm the Earth-Atmosphere system.

**L50** Sedimentation is important for large crystals. In this work it plays a vital roll for the described mechanism. In the revised version, this is explained in several parts of the manuscript, e.g. in the first paragraph of section 3.1. In fact, the oscillations can only take place because of sedimentation; otherwise, the system would fast reach an equilibrium state.

**L79** In the revised version,  $\nabla_x$  is defined in the passage after equation 1.

**L83** We skip this term since it leads to confusion. We investigate an air parcel, i.e. follow the time evolution of the parcel in a Lagrangian way; sedimentation leads to motion of ice crystals relative to the motion of the air parcel.

**L90** Splitting up the velocity into different components allows for separation of the sedimentation term in (new) equation 3 and for changing to the Lagrangian perspective.

**L94 and L101** For the microphysical parameterisations, non-integer moments arise from integrating power-law relationships with non-integer exponents, as used for mass-length-relations or representations of terminal velocities.

**L113** A reference for the two-moment scheme approach is now provided Seifert and Beheng [2006].

**L115** J does not depend on r and is therefore just a factor in the integration w.r.t. r. In the revised version, we left out the sentence because stating that " $\partial J/\partial r = 0$ " is sufficient.

**L159** The geometric standard deviation of the lognormal distribution is dimensionless, it represents the width of the distribution. This has been clarified in the text.

**L158–161** We added a reference for the choice of the settings for solution droplet distribution, motivated by observations.

**L208** We use a simple power law for representing the columnar shape of the ice crystals in a simple way. For this purpose we "fit" this power law (coefficients  $C_i$  and  $\alpha_i$ ) to a more sophisticated relationship from the Spichtinger and Gierens [2009] model.

**L216** We define  $S_i = p_v/p_{si} = \text{RH}_i/100 \%$ . Since  $q_v/q_{v,si} \approx p_v/p_{si}$ , both definitions are almost the same.

**L246** See General comments.

**L261** We now adjusted the spaces between the units.

**L262** For studying the long-term behaviour of SVCs, we have to assure that constant vertical motion is persistent. Slow vertical upward motions  $w < 0.05 \,\mathrm{m \, s^{-1}}$  can be maintained for quite a long time; examples are motions along warm fronts in the extra-tropics or Kelvin waves in the tropics, leading to almost constant vertical velocities over long time. We provided references for these situation. However, the temperature and pressure change in such situations is quite small; this provides a justification for the assumption of constant temperature and pressure, whereas temperature and pressure changes are used in the evolution equation of relative humidity over ice.

**L270-271** The partial derivative term with  $\partial/\partial z$  is a hyperbolic term. See also section 2.2.4, point 3.

L292–294 We rewrote the section and added some more details.

**L296** We referenced the wrong equation, it was supposed to be (37), now (19).

**L299-300** The term  $DEP_{RH}$  is wrong, it has to be  $DEP_q$ , since

$$\left. \frac{\mathrm{d}q_c}{\mathrm{d}t} \right|_{Dep} = \mathrm{DEP}_{\mathrm{q}}.\tag{3}$$

The factor  $\rho$  was also wrong. However, we changed notation so the terms are not called NUC, DEP, SED anymore. Also, we got rid of  $\rho$  in (new) equation (3) using the continuity equation, to avoid confusion.

**L315–318** In the following sections, the abbreviation **F** is used for the right hand side of the system, therefore we decided to already introduce it here.

**L337** Yes, we mean that relative humidity increases again. Sedimentation removes ice crystals, which constitute a sink for relative humidity due to growth.

**L340** State 1 only occurs if either temperature is "rather high" (i.e. right side of the interval  $190 \le T \le 230 \,\mathrm{K}$ ) or the vertical motion is slow (i.e. left side of the interval  $0 < w \le 0.05 \,\mathrm{m \, s^{-1}}$ ). For the qualitative overview we stay at these vague statements, in the bifurcation diagram the quantitative values are given.

**L392** See General comments. The critical point is a stable focus, i.e. a positive (point) attractor.

**L422** The critical point was found by first computing the roots wrt  $RH_i$  of (new) equation (28), former equation (42). From that, the values for  $N_c$  and  $q_c$  were derived analytically using (new) equations (C3) and (C4). See appendix C.

**L439–441** In the revised version, we state that the bifurcation point is a function of w and T.

**L455** Mean sizes are calculated from the mass-size relationship  $L = C_i m^{\alpha_i}$ , as provided in appendix B. Here, we use the mean mass, as given by  $\overline{m} = q_c/N_c$ .

**L462–464** More details on the Spichtinger and Gierens [2009] model are now given in the first paragraph of section 3.6.

**L487** Here we referred to comparison with the model by Spichtinger and Gierens [2009]. We clarified this in the new text.

L502 and L516 "Theoretical" refers to investigations with our simple "analytical" model.

L537 Crystal sizes in this work are to be interpreted as crystal length.

**L610** A more detailed outlook on how minimal models could be useful for cloud parameterisations is given in the conclusions of the revised version.

Figure 7 The median is now indicated by a dotted-dashed line.

**Figure 9** The correct unit is  $m s^{-1}$ .

**Response to Ulrike Wacker (reviewer #2)**

**Theory of dynamical systems**

- We reduced some text about description of SVCs. We added text on the mathematical methods and the conceptual model. We included all indicated references.
- We simplified the system introducing summarising coefficients. The relative humidity is not changed by nucleation, since this constitutes a phase transition from liquid to solid, whereas the gas phase is not changed.

The behaviour of pure ice clouds is completely different than for mixed-phase or pure liquid clouds, which usually exist in a thermodynamic state close to water saturation. Ice clouds are usually not in thermodynamic equilibrium since ice nucleation takes place at high supersaturations and diffusional growth/evaporation is quite slow. Therefore, relative humidity is an important control variable, which is more natural for ice clouds than specific humidity. Since the system does not fulfil mass conservation, there is no additional benefit from changing the variable from  $RH_i$  to  $q_v$  but the formulation of the nucleation rate would be more complicated. Therefore we decided not to transform the system into different coordinates.

- We added a paragraph describing the quality of the different terms in the ODE system.
- We expanded the discussion of the mathematical analysis, including figures of eigenvalues etc. .

**Terminology** We used many terms from theory of dynamical systems from the books by Verhulst [1996] and Argyris et al. [2010]. We tried to change and explain the terms in different ways.

Dry adiabatic lapse rate See general comment.

**Derivation of system of equations** We have shortened the derivation of equations; especially the description of the cloud processes is now partly transferred to appendix B.

**Dimensions** We checked all terms again and corrected the inconsistencies.

**Minor comments**

- We skip the term "Boltzmann equation".
- Equation (9): corrected
- Since the nucleation rate as described by Koop et al. [2000] is formulated as a volume rate, we used this approach. The nucleation rate does not depend on the size of droplets.
- We deleted the term "aerosol particles".
- In contrast to liquid droplet formation the activity is not equal 1 for ice nucleation. The nucleation rate J as described by Koop et al. [2000] can be formulated in terms of differences in water activity (appendix B).
- We changed the layout of equation (16).
- Equation (17) was wrong, now corrected.
- "Length" indicates the length of an ice crystal, i.e. the long side of a columnarshaped ice crystal.

- Droxtal is the term for very small and almost spherical ice crystals.
- We corrected (former) equation (24)
- We corrected (former) equation (30)
- We use latent heat and heat capacity in molar units, i.e.  $L_{ice}$  is actually the molar heat of sublimation; in addition with the molar mass  $M_{air}$  and the molar isobaric heat capacity  $c_p$  the units in the equations are correct. Please note that we also corrected the equation for adiabatic temperature change.
- We moved the respective text.
- Yes, equation (37) is the correct reference.
- $N_a$  is the number concentration of solution droplets per mass dry air. We skip  $n_a$  for clarification.
- We corrected the equation.
- We do not apply w < 0 in our model, only upward vertical motions are considered.
- This is correct, the nucleation rate J is positive for  $RH_i > 100\%$ . Due to the exponential behaviour of J no significant amount of ice crystals is formed at low supersaturations. Only if the supersaturation exceeds certain values (i.e. a "threshold") then a significant amount of ice crystals is produced. We added some text and a reference about the concept of freezing probability and its description by a differential equation, including the nucleation rate.
- We added more text about the use of the Poincaré section for the numerical determination of the limit cycle.
- We rewrote the complete section 3.
- 1. 433: Decreased growth rates lead to slower reduction of supersaturation, thus  $RH_i$  stays longer above the "freezing thresholds" and more ice crystals can be produced.
- We changed the text from "exponential behaviour" to "power law".
- We restructured section 3.
- We added some more text for describing the features of the models by Spichtinger and Gierens [2009] and Kärcher [2002].
- "Analytical model" refers to the model developed in this study.

- Within the first 4 hours of integrations the oscillation is damped and the amplitude in variables decreases. later, the amplitude of the variables increases again. Admittedly, the system does not reach a "constant" limit cycle, since the properties of the limit cycle are continuously changed by cooling of the temperature. We added some text for clarification.
- We added some text about sources, sinks and forcing terms in the beginning of section 3.
- The microphysical properties of the attractors can be similar, since in the limit cycle case the variables changes within a quite large range that includes also the values of the point attractor.
- We arrange the figure captions in a way that they might be understandable without reading the whole text. Thus, some repetition might be possible.
- The dependence of the mean mass on w is only marginal. We added some text in the manuscript to clarify this issue.

**Response to reviewer #3**

**Discussion of SVCs** We enhanced the discussion of SVCs; in fact, we added comparison with remote sensing observations and report more qualitative results of the simulations.

**How realistic are model simulations?**

- Both states can develop, if previous heterogeneous nucleation takes place. In fact, the system will just start at a different point in phase space.
- We have not investigated the system in details for variations in vertical velocity. In fact, it seems from numerical simulations that for weak perturbations, the system approaches similar states as in the undisturbed scenarios. Theoretical analysis would be much more difficult, since we have no longer an autonomous system. The investigation of this system, externally forced by time-dependent vertical velocities is beyond the scope of our study and is dedicated to future work.

**Specific comments**

- 1. Abstract: We changed the text following reviewer's suggestions.
- 2. Introduction: We skip the word "usually".
- 3. Introduction: We added more basic references about heterogeneous nucleation. We also added the suggested references by Immler et al., 2008.
- 4. Section 2, additional settings: We rewrote section 2, changing also the titles of the subsections.

- 5. We added some text about the change in  $n_c = N_c \cdot \rho$  due to temperature/pressure change.
- 6. We defined ODE in the text.
- 7. We rewrote the sentences.
- 8. We added some description of the attractor states, which occur also in the definitions.
- 9. We added the processes.
- 10. We changed the text.
- 11. Former figure 5 (now figure 7) is now described in more details.
- 12. We changed the wording.
- 13. We discussed former figure 6 (now figure 8) in more details. The mean mass is shown in figure 10, thus we do not add another panel to figure 6. We added comparison with remote sensing observations considering the issue of subvisible cirrus (or not).
- 14. In former figure 6 the stable or unstable point attractor is shown. The lines, both solid and dashed, are valid at the respective critical point of the system. The critical point is well defined for both states, it only has different implications, depending on the nature of the state. For the point attractor regime (solid lines), it is a stable focus and the long-term solution approaches it. For the limit cycle regime (dashed lines), the critical point is approximately at the centre of the periodic orbit and provides a good estimate of the mean (or median) properties of the cloud, in terms of mass and number concentration and relative humidity. So the parallel lines in former figure 6, now figure 8, are valid for the critical point which is either a stable focus (solid lines) or an unstable focus (dashed lines).
- 15. We tried to keep the same wording.
- 16. We added some text, see also reply to reviewer #2.

**Comments to figures**

- 1. We changed the caption.
- 2. We changed the caption.
- 3. We changed the figure.
- 4. It is not possible to add other units, since we refer to different temperature, and thus density, regimes. However, for figure 9 showing the number concentrations for point attractor and limit cycle we added units per litre.

- 5. Figure 7 (now figure 9): The solid line represents values at the critical point, regardless of the nature of the critical point. It is well defined for both states, see also comment 14.
- 6. We added some text and we changed the line for the median in the figure. However, the median in the limit cycle case is close to the value at the critical point in the limit cycle regime, not the point attractor. See also comment 14 and comment to figure 6. Limit cycle and point attractor do not coexist, as the environmental conditions determine which state is present.

245 characteristics of the droplets do not depend on the chemical composition. By definition the water activity is the ratio  $e_{sot}/e_{liq}$  of the vapor pressure over a solution,  $e_{sot}$ , and pure liquid water,  $e_{liq}$ . Neglecting the Kelvin effect and assuming that the solution droplets are in equilibrium with the environment ( $e = e_{sot}$ ), the water activity is proportional to the water activity in equilibrium with ice, which is the ratio of the water vapour pressure over ice and pure liquid water:

250
$$a_w = \frac{e_{sol}}{e_{liq}} = \frac{e}{e_{liq}} = \frac{RH_i}{100\%} \frac{e_i}{e_{liq}} = \frac{RH_i}{100\%} a_w^i.$$

Both  $e_i$  and  $e_{liq}$ , only depend on temperature and are parameterised according to Murphy and Koop (2005, eq. 7 and 10, respectively) Hence,  $\Delta a_w$  is a function of  $RH_i$  and T, as given by-

$$\frac{\Delta a_w(T, RH_i)}{= \left(\frac{RH_i}{100\%} - 1\right) a_w^i(T)} = \left(\frac{RH_i}{100\%} - 1\right) \frac{e_i}{e_{iiq}}.$$

255 Therefore J is also a function of  $RH_i$  and T.

The logarithm of the nucleation rate is parameterised by a third order polynomial in  $\Delta a_w$  (Koop et al., 2000, table 1, eq. 7) :

$$\frac{\log_{10} J(T, RH_i) = -906.7 + 8502 \,\Delta a_w}{-26924 (\Delta a_w)^2 + 29180 (\Delta a_w)^3}.$$

260 With this, we can formulate the two terms for particle generation: , similar to the settings by Spichtinger and Gierens (2009) , which are motivated by observations (Minikin et al., 2003) . This leads to a formulation of

$$\underline{NUC_{n}} = \frac{\partial(\rho N_{c})}{\partial t} \frac{\partial N_{c}}{\partial t} \Big| \underbrace{NUC}_{\text{nucleation}} = \frac{4}{3} \pi \underline{\rho} N_{a} \underline{\rho r_{m}^{3}} \exp\left(\frac{1}{2} \left(3 \log \sigma_{r}\right)^{2}\right) J(\underline{RHRH}_{i}, T)$$
(10a)

265 and

$$\underline{NUC_q} \underbrace{\frac{\partial q_c}{\partial t}}_{nucleation} = m_0 \underline{NUC_n} \cdot \underbrace{\frac{\partial N_c}{\partial t}}_{nucleation} |_{nucleation},$$
(10b)

using a typical droplet mean mass  $m_0 = 10^{-15} \text{ kg} (\text{size} \sim 1 \mu \text{m})$  in the spirit of the mean value theorem. The nucleation rate J is parameterised according to Koop et al. (2000) and can be expressed as

270 a function of relative humidity with respect to ice and temperature. For further details see appendix B.

**2.3 Depositional growth**

The growth-

**2.2.1 Diffusional growth**

275 The growth and evaporation of ice crystals is dominated by diffusion . The "advection velocity" g in the mass space is given by of water vapour. With several simplifications of the growth equation for a single ice crystal; this equation has the following form (see, e.g., Stephens, 1983) :-

$$\underline{g(m) = \frac{\mathrm{d}m}{\mathrm{d}t} = 4\pi C D_v^* \rho(q_v - q_{v,si}) f_v}.$$

Here,  $q_{v,s} = \varepsilon p_{si}(T)/p$  denotes the saturation mixing ratio, the shape of the ice crystal is accounted for by the capacity C (assuming the electrostatic analogy, see e.g. McDonald, 1963; Jeffreys, 1918),

 $D_v^*$  is the full diffusion constant including the kinetic correction for small particles (Lamb and Verlinde, 2011) and  $f_v$  denotes the ventilation coefficient.

In this study we make use of the following simplifications: Latent heat release at the crystal surface is neglected and the temperature of the ice particles is assumed to be equal to temperature of ambient

air. We neglect kinetic corrections, since we are mostly interested in growth of larger crystals. Thus,

285

280

$$D_v^* \approx D_v = D_0 \left(\frac{T}{T_0}\right)^{\alpha} \left(\frac{p_0}{p}\right),$$

with  $D_0 = 2.11 \cdot 10^{-5} \text{ m}^2 \text{s}^{-1}$ ,  $T_0 = 273.15 \text{ K}$ ,  $p_0 = 101325 \text{ Pa}$ ,  $\alpha = 1.94$  (e.g. Pruppacher and Klett, 1997). We neglet correction of ventilation, setting  $f_v = 1$ . Since ventilation correction is relevant for very

290 large crystals, this is a reasonable assumption. The shape of ice crystals, is assumed to be prolate spheroids with a length and an eccentricity  $\varepsilon'$ , this leads to the following expression (McDonald, 1963) :

$$C = L \frac{\varepsilon'}{\log\left(\frac{1+\varepsilon'}{1-\varepsilon'}\right)}.$$

we can assume

For the mass-size relation we assume a simple power law  $L(m) = C_i m^{\alpha_i}$  using (for details see appendix B) we obtain the following equation for diffusional growth of a single crystal:

$$g(m) \approx \frac{4}{3} \pi C_i D_v m^{\alpha_i} \rho q_{v,si} (S_i - 1), \tag{11}$$

with constants  $C_i = 1.02 \text{ m}$ ,  $\alpha_i = 0.4$ , which was fitted to the more complex description in Spichtinger and Gierens (2009), where a transition between droxtals and columns is formulated and used.

300

$$\underline{C = \frac{1}{3} C_i m^{\alpha_i}}.$$

Thus, we end with the simplified expression for g(m):

$$\underline{g(m)} = \frac{4}{3}\pi C_i m^{\alpha_i} \rho(q_v - q_{v,si})$$

$$= \frac{4}{3}\pi C_i m^{\alpha_i} \rho q_{v,si}(S_i - 1),$$

305

using the and using saturation ratio  $S_i = p_v/p_{si}$  and the saturation mixing ratio

$$q_{v,si}(T,p) = \frac{\varepsilon p_{si}(T)}{p},$$
(12)

Thus, we can derive the term  $DEP_q$  in equation respectively. We can express the term for diffusional growth in the moment equations (3) by integration, i.e.:

$$\quad \underbrace{\underline{DEP}_{q}}_{q} \underbrace{\frac{\mathrm{d}q_{c}}{\mathrm{d}t}}_{q} \bigg|_{\underbrace{\mathrm{growth}}} = \int_{0}^{\infty} \underline{\rho}g(m)f(m)\,\mathrm{d}m = \frac{4}{3}\pi C_{i}\underbrace{\underline{D}_{v}}_{o}\rho q_{v,si}(S_{i}-1)\mu_{a_{i}}[m] \\ = \frac{4}{3}\pi C_{i}\underbrace{\underline{D}_{v}}_{c}\rho q_{v,si}(S_{i}-1)\underline{\underline{n}}\underbrace{N_{c}^{1-\alpha_{i}}q_{c}^{\alpha_{i}}r_{0}^{\frac{\alpha_{i}(\alpha_{i}-1)}{2}}}_{2}.$$
(13)

**2.3 Sedimentation**

**2.2.1 Sedimentation**

315 For the derivation of the terms  $SED_q$ ,  $SED_n$  we use the mean value theorem for describing the relevant integrals in equation as follows:

$$\int_{0}^{\infty} v_t(m)\rho m^k f(m) \,\mathrm{d}m = \bar{v}_k \int_{0}^{\infty} \rho m^k f(m) \,\mathrm{d}m = \rho \bar{v}_k \mu_k.$$

Thus, we can Following Spichtinger and Gierens (2009), we describe the weighted terminal velocity  $\bar{v}_k$  for the flux of the k-th moment as

320
$$\bar{v}_k = \frac{1}{\mu_k} \int_0^\infty v_t(m) m^k f(m) \, \mathrm{d}m_{\underline{\cdot}},$$
 (14)

(for details see appendix B). Here, we use a simple power law for the representation of the terminal velocity in addition with

$$v_t(m) = \gamma m^{\delta} \operatorname{corr}(T, p) \tag{15}$$

with  $\gamma = 63292.36 \text{ m s}^{-1} \text{ kg}^{-\delta}$ ,  $\delta = 0.57$  and a density correction term e(T,p), i.e.:

325
$$v_t(m) = \gamma m^{\delta} c(T,p)$$

where-

$$c(T,p) = \left(\frac{p}{p_{00}}\right)^{a_i} \left(\frac{T}{T_{00}}\right)^{a_2},$$

 $T_{00} = 233 \text{ K}, p_{00} = 300 \text{ Pa}, a_1 = -0.178, a_2 = -0.397 \text{ (see e.g. Spichtinger and Gierens, 2009) and}$  $\gamma = 63292.36 \text{ ms}^{-1} \text{kg}^{-\delta}, \delta = 0.57$ . Thus, we obtain the weighted velocities for number and mass flux, respectively, in the following form:-

$$\frac{\bar{v}_0}{\underline{v}_0} = \bar{v}_n = \gamma \frac{\mu_\delta}{\mu_0} c(T, p),$$

330

$$\underline{\bar{v}_1} = \overline{v}_q = \gamma \frac{\mu_{\delta+1}}{\mu_1} c(T,p)$$

**$\operatorname{corr}(T, p)$ (see appendix B).**

We can formulate compose the general terms for sedimentation in the moment equations (3):

335
$$\underline{SED_n} = \frac{\partial}{\partial z} \left( \rho \bar{v}_n N_c \right) = \frac{\partial}{\partial z} \left( \rho \gamma \cdot \mu_{\delta+1}[m] c_{\delta}[m] \cdot \operatorname{corr}(T, p) \right),$$
(16a)

$$\underline{\underline{SED}}_{q} = \frac{\partial}{\partial z} \left( \rho \bar{v}_{q} q_{c} \right) = \frac{\partial}{\partial z} \left( \rho \gamma \cdot \mu_{\underline{\delta}}[\underline{m}] \underline{c}_{\underline{\delta}+1}[\underline{m}] \cdot \operatorname{corr}(T, p) \right).$$
(16b)

**2.3 Additional settings**

**2.2.1 Simplifications**

[revised manuscript text omitted]

(29)

where DF|x₀ is the Jacobian of F evaluated at the point x₀. Note that F(x₀) = 0 by definition. The three eigenvalues of the Jacobian, λ₁, λ₂, λ₃, determine the quality of the critical point (Verhulst, 1996, Chapter 3). The eigenvalues must be determined numerically for the relevant parameter values w and T. The Jacobian of the system has two complex conjugate eigenvalues, λ₁ and λ₂λ₁₂ ∈ C, whose real part can be positive or negative, depending on the parameters, w and T. In figure 3 the values of the real part Re(λ₁₂) and the absolute value of the imaginary part [Im(λ₁₂)] are shown.

The third eigenvalue,  $\lambda_3 \in \mathbb{R}$ , is always negative, values are shown in figure 4.

If Complex eigenvalues of the linearised system indicate oscillatory behaviour, which is prevalent in all simulations. As can be seen in figure 3, the real part of the complex conjugate eigenvalues is negative ( $\operatorname{Re}(\lambda_1) = \operatorname{Re}(\lambda_2) < 0$ ), eigenvalues  $\lambda_{1,2}$  can change its sign depending on parameters wand T.

For negative values of the real part ( $\operatorname{Re}(\lambda_{1,2}) < 0$ ) the critical point  $x_0$  is a positive attractorin equation, which means that a solution that starts, i.e. solutions of the ODE (29) starting in a neighbourhood of  $x_0$  will asymptotically converge to  $x_0$  (Verhulst, 1996). This corresponds to state 1, i.e. the damped oscillation where the system reaches stationarity this point approach

[revised manuscript text omitted]

- 690 Figure 8 shows ice crystal mass and number concentrations, respectively, at the critical point,  $x_0$ , as a function of vertical velocity  $(q_c(w), N_c(w))$  for different temperature regimes. The solid lines in both panels correspond to state 1 (attractor regime point attractor regime, damped oscillations), whereas the dashed lines indicate the values at the critical point,  $x_0$ , for state 2 (limit cycle regime, undamped oscillations); note that for state 2,  $x_0$  is an unstable focus.
- Ice crystal number concentrations at the critical point take values in the range  $\frac{3 \times 10^2 \text{ kg}^{-1}}{5} \leq N_c \leq 2 \times 10^5$ 695  $kg^{-1} 3 \times 10^2 kg^{-1} \le N_c \le 2 \times 10^5 kg^{-1}$  (figure 8, top), which corresponds to ice crystal number densities of  $\frac{0.1 \text{L}^{-1} \le n_c \le 110 \text{L}^{-1}}{0.1 \text{L}^{-1} \le n_c \le 110 \text{L}^{-1}}$ . Ice crystal mass concentration ranges between  $4 \times 10^{-9} \le q_c \le 3 \times 10^{-6} \text{ kg kg}^{-1} 4 \times 10^{-9} \le q_c \le 3 \times 10^{-6} \text{ kg kg}^{-1}$  (figure 8, bottom). This corresponds to an ice water content of  $\frac{2.2 \times 10^{-9} \le IWC \le 1.4 \times 10^{-6} \text{ kgm}^{-3} 2.2 \times 10^{-9} \le IWC \le 1.4 \times 10^{-6} \text{ kgm}^{-3} 2.2 \times 10^{-9} \le IWC \le 1.4 \times 10^{-6} \text{ kgm}^{-3} 2.2 \times 10^{-9} \le IWC \le 1.4 \times 10^{-6} \text{ kgm}^{-3} 2.2 \times 10^{-9} \le IWC \le 1.4 \times 10^{-6} \text{ kgm}^{-3} 2.2 \times 10^{-9} \le IWC \le 1.4 \times 10^{-6} \text{ kgm}^{-3} 2.2 \times 10^{-9} \le IWC \le 1.4 \times 10^{-6} \text{ kgm}^{-3} 2.2 \times 10^{-9} \le IWC \le 1.4 \times 10^{-6} \text{ kgm}^{-3} 2.2 \times 10^{-9} \le IWC \le 1.4 \times 10^{-6} \text{ kgm}^{-3} 2.2 \times 10^{-9} \le IWC \le 1.4 \times 10^{-6} \text{ kgm}^{-3} 2.2 \times 10^{-9} \le IWC \le 1.4 \times 10^{-6} \text{ kgm}^{-3} 2.2 \times 10^{-9} \le IWC \le 1.4 \times 10^{-6} \text{ kgm}^{-3} 2.2 \times 10^{-9} \le IWC \le 1.4 \times 10^{-6} \text{ kgm}^{-3} 2.2 \times 10^{-9} \le IWC \le 1.4 \times 10^{-6} \text{ kgm}^{-3} 2.2 \times 10^{-9} \le IWC \le 1.4 \times 10^{-6} \text{ kgm}^{-3} 2.2 \times 10^{-9} \le IWC \le 1.4 \times 10^{-6} \text{ kgm}^{-3} 2.2 \times 10^{-9} \le IWC \le 1.4 \times 10^{-6} \text{ kgm}^{-3} 2.2 \times 10^{-9} \le IWC \le 1.4 \times 10^{-6} \text{ kgm}^{-3} 2.2 \times 10^{-9} \le IWC \le 1.4 \times 10^{-6} \text{ kgm}^{-3} 2.2 \times 10^{-9} \le IWC \le 1.4 \times 10^{-6} \text{ kgm}^{-3} 2.2 \times 10^{-9} \le IWC \le 1.4 \times 10^{-6} \text{ kgm}^{-3} 2.2 \times 10^{-9} \le IWC \le 1.4 \times 10^{-6} \text{ kgm}^{-3} 2.2 \times 10^{-9} \le IWC \le 1.4 \times 10^{-6} \text{ kgm}^{-3} 2.2 \times 10^{-9} \le IWC \le 1.4 \times 10^{-6} \text{ kgm}^{-3} 2.2 \times 10^{-9} \le IWC \le 1.4 \times 10^{-6} \text{ kgm}^{-3} 2.2 \times 10^{-9} \le IWC \le 1.4 \times 10^{-6} \text{ kgm}^{-3} 2.2 \times 10^{-9} \le IWC \le 1.4 \times 10^{-6} \text{ kgm}^{-3} 2.2 \times 10^{-9} \le IWC \le 1.4 \times 10^{-6} \text{ kgm}^{-3} 2.2 \times 10^{-9} \le 10^{-9} \text{ kgm}^{-3} 2.2 \times 10^{-9} \text{ kgm}^{-3} 2.2 \times 10^{-9} \text{ kgm}^{-3} 2.2 \times 10^{-9} \text{ kgm}^{-3} 1.2 \times 10^{-9} 1.2 \times 10^{-9} \text{ kgm}^{-3} 1.2 \times 10^{-9} \text{ kgm}^{-3} 1.2 \times 10^{-9} \text{ kgm}^{-3} 1.2 \times 10^{-9} 1.2 \times 10^{-9} 1.2 \times 10^{-9} 1.2 \times 10^{-9} 1.2 \times$

[revised manuscript text omitted]

---

## Referee Report (RR1)

*2nd review of*
**Subvisible cirrus clouds – a dynamical system approach; revised version**
*by E.J. Spreitzer et al.*

**Summary:**
As stated in my first review, I appreciate the research presented in the paper. The revised version is definitely improved. In particular it benefits from, e.g., the shift of microphysical details and derivations to the appendices. And I want to higlight the presentation of the parameterization idea in the Conclusions.
Before publication, I suggest a straightforward amendment, including
to present the paper concisely ('Conclusions' are ok), to call the same thing everywhere by the same technical term, to arrange clearly the equs., to inspect the equs carefully, to avoid repetitions, and to skip material that is really basic knowledge or not absolutely necessary.
A list of specific examples (not complete) is given below.
After the 1st revision, this should not pose a severe problem anymore.

**Specific comments**

1. The terms equilibrium state, critical state, point attractor, singular point are used alternately, e.g. l. 81-85, l.410-412, Reply to Rev.2 p.2, and others. Please be more precise, although you find the mismatch in some textbooks. In your model, the thermodynamic equilibrium requires ice saturation, but this needs $w = 0$. For $w > 0$, you examine an open system and you look for steady states (which you call 'critical points', although this expression can be misleading). The term 'attractor' implies that the longterm behaviour of the system will end in a part of the phase space (e.g., point a., periodic a.); the opposite is a repellor. I have the impression that you use 'positive a.' for a stable node or focus.
   Please screen the manuscript for consistency and keep track with the wording.

2. For the final version, please focus the descriptions in Section 2 - in combination with App. A, B. Few examples: App. A does not contain anything else but given in Section 2 together with the very basic principle of total mass conservation. Shorten l.218-224 to one line. Shorten l. 241-246 to something like '$w$, $T$, and $p$ are assumed constant and treated as control parameters.' Para l.252-255 is mostly a repetition, same for l.270-272. l.247-249 Sentence 'If ...' is redundant.
   The transformation rates can be written in the same formal structure to simplify the reading and retracing. E.g., the growth term (13), (25b), (B9), and coefficient (B13d) should have the same structure and variables; do not switch between $RH_i$, $S_i$, $p_v$, and $q_v$ as well as $p_{si}$ and $q_{si}$ etc. if not absolutely necessary.
   Please go through all equations.

3. l. 68/69. You 'exclude the possibility of liquid origin ice clouds'. Isn't this a contradiction to your response to reviewer 2, 2nd comment, that nucleation of ice particles is from the LIQUID phase? Please clarify.

4. l. 95. 'system variable' instead 'control v.'.

5. l.139. Skip 'dry'.

6. l. 161p. You write that $f_a$ were normalized, but this neither coincides with the unit of $f_a$ nor with e.g., Eq.(8). Please clarify.

7. Eq.(B13a) does not match with (9) and (10a). Please check.

8. l. 275, Eq.(21): The factor $\partial RH_i/\partial q_v$ is missing on the RHS.
   Please check in (B13e) whether a factor $\varepsilon$ is missing in the first bracketed term.

9. l. 279pp and your reply to Rev.2. Mathematically, both variables $RH$ and $q_v$ can be used as prognostic variable, since both carry the information on the amount of water vapour. We should stop the discussion, although I do not follow your arguments.
   Pure ice clouds can indeed exist close to thermodynamic equilibrium (ice saturation), the single steady state in case of $w \to 0$.

10. Eq.(25b). The sedimentation term should be proportional to $N_c^{-\delta} q_c^{\delta+1}$ (instead of $N_c^{-\delta} q_c^{\delta-1}$). Please check.

11. p.25 Eq. (B13b,c): The variable $c$ occurs in 2 different meanings, $c$ and $c(T,p)$. Please distinguish. Eq.(16), (34), (B13). Please unify corr(T,p) and c(T,p).

12. After Eq.(25). The new presentation is definitely improved. Unfortunately, only much later (l.373pp) you interpret the terms as internal transformation rates and external sources/sinks.
   Please work out sedimentation as external sink, the $w$-term as external source, and the nucleation terms also as external sources (since ice particles form from an inexhaustable external reservoir of liquid droplets). Only the 'diffusional growth' is an internal transformation. This can be seen seen from Eq.(25) (and more easily if $q_v$ is prognostic variable), hence please consider to place the discussion there or in the initial part of section 3.3.
   l. 332-334. I do not see this. Without sedimentation, only external SOURCES exist, i.e. nucleation and the $w$ effect. For $RH_i = 100\%$, $dRH_i/dt > 0$, that is no steady state.

13. l. 376 'artificially produced vapour'. Please reformulate, since vapour enters your system as external source.

14. l. 380-407, 'dissipative system'. The previous text explains the externally forced system, not ('Thus'?) its dissipative character. The discussion of $\nabla \cdot \mathbf{F}$ (27) is on the contribution of each term, not on 'dissipative system'. What is the aim? A dissipative system is characterized by the negative value of the temporal average of $\nabla \cdot \mathbf{F}$ in its longterm behaviour, not necessarily at each instance. I wonder whether you need this passage.

15. l.438-444. This is an important point. If only a single steady state exists and if this is stable, ALL trajectories should end up there. Focus on this aspect and skip talking about trajectories starting in the neighborhood for conciseness.

16. l.461-471 Poincare-map. Although this is a nice calculation, I do not see any added value, since you have found and discussed the limit cycle before (figs. 2, 6). If it is for the purpose to calculate the period (l.534p), please tell so and skip the rest.

17. l.588 'analytical model', fig. 13,14 'simple model' and perhaps other locations. You talk about the model presented in this paper. It is not an anaytical model, since you solve most of the equs. numerically. And it is by no means a 'simple' model. Please choose a proper name and use this name throughout the paper.

18. l. 648. Which microphysical properties? Maybe you address number and mass concentrations, but these are no 'microphysical' properties.

19. l. 650. 'in the point attractor case'.

20. Figs. 13, 14 are a repetition of Figs. 1, 2, supplemented by other model results. Figs. 13, 14 can replace Figs. 1,2 for shortness.

---

## Referee Report (RR2)

*3rd review of*
**Subvisible cirrus clouds – a dynamical system approach; revised version**
*by E.J. Spreitzer et al.*

**Summary:**
As stated in my previous reviews, I appreciate the research presented in the paper. The revised version is improved.
We may continue a lengthy discussion on several passages. However, the authors are ultimately responsible for their paper.
I have few minor points for corrections, listed below, and also I recommend the authors to go through the text again to avoid repetitions.
Apart from that, I recommend publication.

**Few specific comments**

1. Point 6 from previous review, "normalisation of $f_a$", l. 179: You introduce $f$, which fulfills $\int f \, dm = N_c$ without further saying. Matters are equally straightforward for $f_a$, that is $\int f_a \, dr = N_a$. Why make things more complicated?

2. Previous point 11. The correction factor is not unified everywhere.
$corr(T,p)$ occurs in, e.g. (A12), but not in (A13).
Eqs.(A13c) contains a $c$-variable twice, $c$ and $c(T,P)$. Please clarify.

3. l. 147. Something is strange in the typesetting of the limiting values for $f(m)$.

4. l. 415. The previous sentence explains the presence of an externally forced system. The sentence starting with 'Thus' is a repetition of the previous one. When using the word 'dissipative', then please explain why the dissipative nature follows from the previous statement.
See again previous point 14.

5. l. 420. 'contraction or expansion of system solutions' is not a good expression.

6. l. 474, 475. Use '.... state $\mathbf{x_0}$ is stable.'

7. l. 475, 490. Apart from these two lines (and figure captions) 'source' and 'sink' are used in the sense of (e.g., mass) source and sink of the system. Now you introduce the same words in a different context/meaning; this is confusing. The passages can be skipped without loss of information.

8. l. 510. Avoid 'positive attractor'.

---

## Author Response (AR2)

**Reply to Ulrike Wacker's comments**

We thank Ulrike Wacker for her careful and important comments. As in the first round of reviews, these comments helped to improve the manuscript's quality.

In the following we give a detailed reply to all comments:

1. We changed the terminology for the whole manuscript, using equilibrium state instead of critical points etc. For this purpose we used the wording from the standard reference Hirsch et al. (2013) as suggested by the editor.

2. We skipped Appendix A and shortened the description of section 2. The presentation of coefficients has been refined, using the same structures and variables.

3. The terminology is a bit misleading, but unfortunately these terms were already introduced to the community by Krämer et al. (2016). For clarification, we added some words, here is a short summary for the two different states:

   - liquid origin ice formation: Freezing of pre-existing cloud droplets at thermodynamica states close to water saturation and at temperatures $T \geq 235\,\mathrm{K}$.
   - in situ ice formation: Nucleation of ice crystals from solid aerosols (heterogeneous nucleation, as e.g. deposition nucleation) or from supercooled solution droplets at thermodynamic states far below water saturation (but ice supersaturation) and temperatures $T < 235\,\mathrm{K}$.

   For details see also explanation in Wernli et al. (2016). We exclude liquid origin ice crystals, since we are interested in ice formation in slow large-scale updraughts at low temperatures $T < 235\,\mathrm{K}$.

4. Done.

5. Sentence has been reformulated.

6. The units of $f_a$ are $[f_a] = \mathrm{kg}^{-1}\,\mathrm{m}^{-1}$, $f_a$ is normalized to the total number concentration of aerosols.

7. We have corrected the equations, now they are correct and consistent.

8. No, the factor is not missing, since $\frac{\partial RH_i}{\partial q_v} = \frac{RH_i}{q_v} = \frac{100\%}{q_{v,si}}$. We clarified this in the text.

9. We added some words for clarification.

10. Correct.

11. We unified the representation of the correction factor.

12. We added some text for clarification and changed the description according to the suggestion.

13. Done.

14. We use this analysis as a first qualitative description; we removed the parts about the dissipative character of the system, since we are interested in the time evolution in terms of expansion/contraction of the system.

15. We keep the first part with the correct description of stability using neighbourhoods, since this is the mathematically correct formulation. However, we added a sentence following the reviewer's suggestion.

16. The Poincare map is used for the numerical determination of the limit cycle, thus we keep this part in the manuscript.

17. We have chosen the term "reduced model".

18. We changed the description.

19. Changed.

20. We keep the figures as they are, since the additional curves at the beginning would confuse the reader.

[revised manuscript text omitted]

---

## Author Response (AR3)

**Reply to Ulrike Wacker's comments**

22 May 2017

**Response to Ulrike Wacker (reviewer #2)**

We thank Ulrike Wacker for her comments, helping to improve our manuscript.

1. We rewrote the text for clarifying the normalisation for both distributions $f$ and $f_a$ in a consistent way.

2. Corrected

3. We changed the typesetting of the boundary condition as follows:

$$\lim_{m \to \infty} f(m, \vec{x}, t) = 0$$

4. We rewrote the text to clarify our analysis.

5. This expression was deleted.

6. We followed the suggestion of the reviewer.

7. We skipped the passages as suggested.

8. We wrote "stable attractor" instead.

[revised manuscript text omitted]